

**Rapid environmental responses to climate-induced hydrographic changes in**
**the Baltic Sea entrance**
LAURIE M. CHARRIEAU[1], KARL LJUNG[1], FREDERIK SCHENK[2], UTE DAEWEL[3], EMMA
KRITZBERG[4] and HELENA L. FILIPSSON[*][1]
[1]Department of Geology, Lund University, Sweden
[2]Bolin Centre for Climate Research and Department of Geological Sciences, Stockholm University, Sweden
[3]Department of System Analysis and Modelling, Centre for Materials and Coastal Research, Geesthacht,
Germany
[4]Department of Biology, Lund University, Sweden
[*]Corresponding author (address: Sölvegatan 12, SE-223 62; e-mail: helena.filipsson@geol.lu.se)
Key-words: benthic foraminiferal; NAO index; environmental reconstruction; Anthropocene;
Öresund
Abstract
The Öresund (the Sound), which is a part of the Danish straits, is linking the marine North Sea
and the brackish Baltic Sea. It is a transition zone where ecosystems are subjected to large
gradients in terms of salinity, temperature, carbonate chemistry, and dissolved oxygen
concentration. In addition to the highly variable environmental conditions, the area is responding
to anthropogenic disturbances in e.g. nutrient loading, temperature, and pH. We have
reconstructed environmental changes in the Öresund during the last c. 200 years, and especially
dissolved oxygen concentration, salinity, organic matter content, and pollution levels, using
benthic foraminifera and sediment geochemistry. Five zones with characteristic foraminiferal
assemblages were identified, each reflecting the environmental conditions for respective period.



The largest changes occurred ~1950, when the foraminiferal assemblage shifted from a low
diversity fauna, dominated by the species *Stainforthia fusiformis* to higher diversity and
abundance, and dominance of the *Elphidium* group. Concurrently, the grain-size distribution
shifted from clayey — to more sandy sediment. To explore the causes for the environmental
changes, we used time-series of reconstructed wind conditions coupled with large-scale climate
variations as recorded by the NAO index, as well as the ECOSMO II model of currents in the
Öresund area. The results indicate increased changes in the water circulation towards stronger
currents in the area since the 1950's. The foraminiferal fauna responded quickly (< 10 years) to
the environmental changes. Notably, when the wind conditions, and thereby the current system,
returned in the 1980's to the previous pattern, the foraminiferal species assemblage did not
rebound, but the foraminiferal faunas rather displayed a new equilibrium state.
1 – Introduction
The Öresund (the Sound) is one part of the Danish straits between Sweden and Denmark.
Together with the Great — and Little Belt, they link the open-ocean waters of the North Sea and
the brackish waters of the Baltic Sea. The confluence of the water masses creates a north-south
gradient as well as a strong vertical stratification of the water in terms of salinity, carbonate
chemistry and dissolved oxygen concentration ($[O_2]$) (Leppäranta and Myrberg 2009). The depth
of the halocline mainly depends of the outflows from the Baltic Sea; a strong thermocline
develops during spring and summer, which further strengthens the vertical stratification. Thus,
the ecosystems in the Öresund are exposed — and adapted — to a unique transitional
environment. The region is also characterized by intense human activities, with 4 million people
living in the vicinity of the Öresund and 85 million people living in the catchment area of the
Baltic Sea. Discharge from agriculture, industry, and urban areas on both the Swedish and



Danish sides of the strait, and the considerable impact of marine traffic – the strait is one of the
busiest waterways in the world – generate pollution and eutrophication of the water (HELCOM
2009; ICES 2010). Since the 1980's, the implementation of efficient wastewater treatment and
measures in agriculture contributed to markedly reduce the amount of nutrients coming from
river run-off (Nausch et al. 1999; Carstensen et al. 2006; Rydberg et al. 2006). However, these
efforts in decreasing nutrient loads have not resulted in improved water quality, due to the long
time scales of biogeochemical cycles to reach equilibrium in the Baltic Sea region (Gustafsson et
al. 2012). The Öresund, like most of the Baltic Sea, is still assessed to be eutrophic, and hypoxic
events are frequent (Rosenberg et al. 1996; Conley et al. 2007, 2011; HELCOM 2009;
Wesslander et al. 2016). Moreover, increasing temperatures and declining pH, linked to global
climate change and ocean acidification, have been reported for surface and bottom waters in the
area (Andersson et al. 2008; Göransson 2017). As a result, ecosystems in the Öresund are
currently under the combined impact of natural and anthropogenic stressors (Henriksson 1969;
Göransson et al. 2002; HELCOM 2009; ICES 2010). The multiple stressors currently affecting
the environment make this region particularly interesting to study, and also highlight the need to
obtain records of decadal and centennial environmental changes. As noted above, both recent
human-induced impacts and climate variability have been substantial in the region. Therefore the
question arises whether these factors have affected the benthic environment. Furthermore,
sediment records of past environmental changes can provide crucial context for ongoing and
future predicted changes in the Öresund and Baltic Sea regions.
We used the marine sediment record and its contents of foraminifera as well as sediment
geochemistry to obtain records of decadal environmental changes. Benthic foraminifera are
widely used for environmental reconstructions, based on their rapid response to environmental



changes, broad distribution, high densities, and often well-preserved tests (shells) in the sediment
(e.g. Sen Gupta 1999). For instance, distribution of benthic foraminifera have been used for
historical environmental reconstructions of fjords on decadal to centennial timescales on the
Swedish west coast (Nordberg et al. 2000; Filipsson and Nordberg 2004a, 2004b; Polovodova
Asteman and Nordberg 2013; Polovodova Asteman et al. 2015), and in the Kattegat
(Seidenkrantz 1993; Christiansen et al. 1996). In the Öresund, living foraminiferal assemblages
have been studied (Hansen 1965; Charrieau et al. 2018), but to the best of our knowledge, no
studies of past foraminiferal assemblages have been performed. In this study, we used
foraminiferal fauna analysis in combination with sediment geochemistry and grain-size analyses
to reconstruct the environmental conditions of benthic systems during the last two centuries in
the Öresund. Furthermore, we analyzed long time series of wind conditions in the area to
evaluate the coupling between local changes in ecosystem variables and variations in
atmospheric and subsequent hydrographic conditions, and a possible link with large-scale
variations expressed through the North Atlantic Oscillation (NAO) index. Finally, we compared
our data with the model ECOSMO II (Daewel and Schrum 2013; 2017) of currents and water
circulation changes in the Öresund area during the period 1948—2013.
2 – Study site
The Öresund is a 118 km long narrow strait (Figure 1). The water depth in the northern part is on
average 24 m but it reaches 53 m south of the Island of Ven. The Öresund is an important link
between the North Sea, Skagerrak, Kattegat and the Baltic Sea (Figure 1), and up to 30 % of the
water exchange in the region goes through the Öresund (Sayin and Krauß 1996; Leppäranta and
Myrberg 2009); the remaining part goes through the Great and Little Belt. The width of the
Öresund varies between 4 and 28 km, and the water has overall high current velocities, up to 1.5



m.s$^{-1}$ at the upper water layer in the northern part (Nielsen 2001). The fully marine Skagerrak
consists of water masses from the North Sea and the North Atlantic and in general a thin surface
layer with water originating from the Baltic Sea and rivers draining into the sea; the water
circulation forms a cyclonic gyre (cf. Erbs-Hansen et al. 2012). Part of the Skagerrak waters
reach the Kattegat and the Baltic Sea, where they are successively diluted with the large amounts
of freshwater (around 15,000 m$^3$/s, Bergström and Carlsson 1994) draining into the Baltic Sea
from numerous large rivers. The low-saline Baltic Sea surface water is transported by the Baltic
Current, which is typically confined along the Swedish west coast in the Kattegat but may cover
a larger surface area towards the west, depending on wind direction. The Baltic Current later
joins the Norwegian Coastal Current in the Skagerrak (Figure 1). The large fresh water input and
the subsequent large salinity difference between the Kattegat and Baltic Sea result in a two-layer
structure in the Öresund (Figure 2) (She et al. 2007; Leppäranta and Myrberg 2009). The water
stratification is influenced by the surface water from Arkona Basin (salinity 7.5—8.5), the
surface water from the Kattegat upper layer (salinity 18—26) and the lower layer of the Kattegat
(salinity 32—34).
Salinity, temperature, pH, [O$_2$] and nutrient content, here represented by dissolved inorganic
nitrogen concentration [DIN] (nitrate + nitrite + ammonium), in the surface and bottom waters of
the Öresund vary seasonally (Figure 3, Appendix A). In the surface and bottom water, salinity
ranges between ~8 and ~18 and between ~29 and ~34, respectively, and it is more stable between
April and July, when the stratification is the strongest (Figure 3). Temperature ranges between
~1 °C in February and ~19 °C in July in the surface water, while in the bottom water, the lowest
temperature is found in March—April with ~5° C, and the highest temperature in October—
November with ~13 °C. The pH varies between ~8.1 and ~8.6 in the surface water, and between



~7.8 and ~8.6 in the bottom water, without a clear seasonal pattern (Figure 3). [O$_2$] in the bottom
water reaches ~7 mL.L$^{-1}$ in January, and it is typically below 2 mL.L$^{-1}$ in October, approaching
hypoxic values. In the surface water, [DIN] can reach ~7 µmol.L$^{-1}$ in January, and it is ~0
µmol.L$^{-1}$ between April and August (Figure 3).
3 - Materials and Methods
3.1 Sampling
A suite of sediment cores, as well as water samples from the water column, were collected in
November 2013 during a cruise with r/v *Skagerak*. Here we present the data from two sediment
cores sampled at the Öresund station DV-1 (55°55.59' N, 12°42.66' E) (Figure 1), north of the
Island of Ven. The water depth was 45 m, and CTD casts were taken to measure salinity,
temperature and [O$_2$] in the water column. Water samples were collected at 10, 15, 20, 30 and 43
m from the Niskin bottles for carbonate chemistry analyses. The CTD and carbonate chemistry
data are presented in Charrieau et al. (2018). The salinity profile in the water column showed the
typical halocline at 10 m depth (Figure 2). The temperature and [O$_2$] decreased with depth. The
pH values decreased with depth and increase again when reaching the bottom water (Figure 2).
In general, it is challenging to obtain sediment cores in the Öresund, due the high current
velocities up to 1.5 m.s$^{-1}$ (Nielsen 2001), human-induced disturbances, and limited areas of
recent sediment deposition (Lumborg 2005), but our site north of Ven represents an
accumulation area. The cores (9-cm-inner-diameter) were collected using a GEMAX twin barrel
corer. The corer allowed sampling of 30 and 36 cm long sediment cores (referred in this study as
core DV1-G and DV1-I, respectively), which were sliced into one centimeter sections. The
samples from the DV1-G core were analyzed for carbon and nitrogen content, grain size



distribution, and dated using Gamma spectroscopy. The samples from the DV1-I core were
analyzed with respect to foraminiferal fauna and carbon and nitrogen content. The distinct
carbon content profiles, measured on both cores, were used to correlate the $^{210}$Pb dated DV1-G
core to the DV1-I core used for foraminiferal analyses.
3.2 Chronology
The age-depth model was established using $^{210}$Pb and $^{137}$Cs techniques on samples from the
DV1-G core. The samples were measured with an ORTEC HPGe (High-Purity Germanium)
Gamma Detector at the Department of Geology at Lund University, Sweden. Corrections for
self-absorption were made for $^{210}$Pb following Cutshall et al. (1983). The instruments were
calibrated against in-house standards and the maximum error was 0.5 year in the measurements.
Excess (unsupported) $^{210}$Pb was measured down to 23 cm and the age model was calculated
based on the Constant Rate of $^{210}$Pb Supply (CRS) model (Appleby 2001).
3.3 Foraminifera analyses
The foraminiferal samples were prepared following standard micropalaeontological techniques
(e.g. Murray 2006). Approximately 10 g of freeze-dried sediment per sample were wet sieved
thought a 63-μm mesh screen and dried on filter paper at room temperature. Subsequently, the
samples were dried sieved through 100- and 500-μm mesh screens and separated into the
fractions 100-500 μm and >500 μm. The foraminifera from every second centimeter of the core -
plus from additional centimeters around key zones - were picked and sorted under a Nikon
microscope. A minimum of 300 specimens per sample were picked and identified, if necessary
the samples were split with an Otto splitter (Otto 1933). For taxonomy at the genus level, we
mainly followed Loeblich and Tappan (1964) with some updates from more recent literature, e.g.



Tappan and Loeblich (1988). For taxonomy at the species level, we mainly used Feyling-
Hanssen (1964), Feyling-Hanssen et al. (1971) and Murray and Alve (2011). For original
descriptions of the species, see Ellis and Messina (1940 and supplements up to 2013).
Recently, the eastern Pacific morphospecies *Nonionella stella* has been presented as an invasive
species in the Skagerrak-Kattegat region (Polovodova Asteman and Schönfeld 2015). However,
a comparison of *N. stella* DNA sequences from the Santa Barbara Basin (USA) (Bernhard et al.
1997) with the Swedish west coast specimens demonstrates that they represent two closely
related species but are not conspecific (Deldicq et al. in press). Therefore, we have referred to the
species found here as *Nonionella* sp. T1, following Deldicq et al. (in press). The species
*Verneuilina media* (here referred to the genus *Eggerelloides*), which has often been reported in
previous studies from the Skagerrak-Kattegat area (e.g. Conradsen et al. 1994), was
morphologically close to *Eggerelloides scabrus* in the present material, and these two species
have been grouped as *E. medius/scabrus*. The taxon *Elphidium excavatum* forma *clavata* (cf.
Feyling-Hanssen 1972), was referred to as *Elphidium clavatum* following Darling et al. (2016).
*Elphidium clavatum* and *Elphidium selseyense* (Heron-Allen and Earland) were morphologically
difficult to separate in this region, as transitional forms occur. The dominant species was *E.*
*clavatum*, but we acknowledge that a few individuals of *E. selseyense* could have been included
in the counts. The taxon *Ammonia beccarii* was referred to as *Ammonia batava*, following recent
molecular work done on the taxon *Ammonia* in the Kattegat region (Groeneveld et al., 2018; Bird
et al. in press).
Foraminiferal density was calculated and normalized to the number of specimens per 50 cm$^3$.
Data of densities of living + dead foraminifera for the first two centimeters of the core are from
Charrieau et al. (2018). Some specimens displayed decalcified tests, however the inner organic



linings were preserved. These inner organic linings were reported separately and not included in
the total foraminiferal counts. Benthic foraminiferal accumulation rates were calculated as
follows:
BFAR (number of specimens.cm$^{-2}$.yr$^{-1}$) = BF x SAR,
where BF is the number of benthic foraminifera per cm$^3$ and SAR is the sediment accumulation
rate (cm.yr$^{-1}$). Foraminiferal species that accounted for >5 % of the total fauna in at least one of
the samples were considered as major species, and their density was used in statistical analysis.
To determine foraminiferal zones, stratigraphically constrained cluster analysis was performed,
using the size-independent Morisita's index to account for the large differences in the densities
between samples. A dendrogram was then constructed based on arithmetic averages with the
UPGMA method (Unweighted Pair Group Method with Arithmetic Mean). Correspondence
analysis was also performed, to determine significant foraminiferal species in each zone.
Statistical analyses were performed using the PAST software (Hammer et al. 2001).
3.4 Organic matter analyses
Total Organic Carbon (TOC) and Total Nitrogen (TN) content were measured for both DV1-G
and DV1-I. Approximately 8 mg of freeze-dried sediment was homogenized for each centimeter
and placed in silver capsules. Removal of inorganic carbon was carried out by in-situ
acidification (2M HCl) method based on Brodie et al. (2011). TOC and TN content were
analyzed on a Costech ECS 4010 Elemental Analyzer at the Department of Geology, Lund
University. The instrument was calibrated against in-house standards. The analytical precisions
showed a reproducibility of 0.2 % and 0.03 % for TOC and TN contents, respectively. The molar
C/N ratio was calculated.





3.5 Grain-size analyses
Grain-size analyses were performed on core DV1-G using 3.5 to 5 g of freeze-dried sediment for
each centimeter. Organic matter was removed by adding 15 mL of 30 % $H_2O_2$ and heating
during 3 to 4 minutes until the reaction ceased. After the samples had cooled down, 10 mL of
10 % HCl was added to remove carbonates; thereafter the sediment was washed with milli-Q
until its pH was neutral. In the last step, biogenic silica was removed by boiling the sediment in
100 mL 8 % NaOH, and then washed until neutral pH was reached. The sand fraction (>63 µm)
was separated by sieving and the mass fraction of sand of each sample was calculated. Grain
sizes <63 µm were analyzed by laser diffraction using a Sedigraph III Particle Size Analyzer at
the Department of Geology, Lund University. The data were categorized into three size groups,
<4 µm (clay), 4–63 µm (silt) and 63–2000 µm (sand).
3.6 Climate data and numerical modeling
Data from the dataset High Resolution Atmospheric Forcing Fields (HiResAFF) covering the
time period 1850–2008 (Schenk and Zorita 2012; Schenk 2015) were used to study the variations
of near-surface (10 m) wind conditions during the winter half of the year (October to March).
The daily dataset can be downloaded from WDC Climate (Schenk 2017). Wind conditions over
the Öresund are represented by the closest grid point of HiResAFF at 55° N and 12.5° E. The
NAO index as defined by Jones et al. (1997) for boreal winter (December to March) was used,
with updates taken from the Climate Research Unit (CRU,
https://crudata.uea.ac.uk/cru/data/nao/). To allow comparison, the NAO and wind data were
normalized relative to the period 1850–2008. Changes in the currents through the Öresund and
the Kattegat were taken from the fully coupled physical biogeochemical model ECOSMO II





(Daewel and Schrum 2013, 2017), which was forced by NCEP/NCAR reanalysis data and covers
the period 1950–2013. On model ECOSMO II, the simulated South-North currents are
represented as VAV (vertically averaged V- component) and the simulated West-East currents as
UAV (vertically averaged U - component).
4 – Results
4.1 Age model
The unsupported $^{210}$Pb showed a decreasing trend with depth in the DV1-G core (Figures 4A,
4B). The peak observed in the $^{137}$Cs around 9 cm corresponds to the Chernobyl accident in 1986
(Figure 4C). The unsupported $^{210}$Pb allowed direct dating of the core between 2013 and 1913.
The sedimentation rate ranged between 1 and 5.6 mm.y$^{-1}$, with an average of 2.2 mm.y$^{-1}$, and
was decreased with depth. The ages of the lower part of the sediment record were deduced by
linear extrapolation based on a sedimentation rate of 1.4 mm.y$^{-1}$, corresponding to the linear
mean sedimentation rate between the years 1913 and 1946 (Figure 4D).
4.2 Foraminiferal assemblages and sediment features
The foraminiferal assemblages were composed of 76 species from the porcelaneous, hyalines
and agglutinated forms (Appendix B). Eleven foraminiferal species had relative abundance
higher than 5 % in at least one sample and were considered as major species (Plate 1, Figure 5).
The cluster analysis reveals three main foraminiferal zones (FOR-A, FOR-B, and FOR-C),
separated into five subzones to which we assigned dates according to the age model: FOR-A1
(1807–1870), FOR-A2 (1870–1953), FOR-B1 (1953–1998), FOR-B2 (1998–2009), and FOR-C
(2009–2013) (Figures 5, 6). The correspondence analysis resulted in three factors explaining



92 % of the variance, and in assemblages consisting in seven significant species, presented in
order of contribution: *Nonionella* sp. T1, *Nonionoides turgida*, *Ammonia batava*, *Stainforthia*
*fusiformis*, *Elphidium albiumbilicatum*, *E. clavatum* and *Elphidium magellanicum* (Table 1).

421. Zone FOR-A1 (1807–1870)

The foraminiferal accumulation rate (BFAR) was on average 5 ±3 specimens.cm$^{-2}$.y$^{-1}$ in zone
FOR-A1 (Figure 5). The Shannon index was stable and low, around 1.77 ±0.1 (Figure 5). The
agglutinated species *Eggerelloides medius/scabrus* and the hyaline species *Stainforthia*
*fusiformis* made major contributions to the assemblages (relative abundances up to 53 % and
34 %, respectively; Figure 5A). *Ammonia batava*, the three *Elphidium* species (*E.*
*albiumbilicatum*, *E. clavatum*, and *E. magellanicum*), *Nonionellina labradorica* and the
agglutinated species *Reophax subfusiformis* were also major species with abundances up to 7 %.
The TOC and C/N values on this period were stable and were on average 3.36 % and 8.8 %,
respectively (Figure 7). The clay size fraction dominated the sediment at the end of this period
with a mean value of 63 %, and the sand content was around 7 % (Figure 7).

422. Zone FOR-A2 (1870–1953)

The BFAR was on average 9 ±5 specimens.cm$^{-2}$.y$^{-1}$ in zone FOR-A2 (Figure 5). The Shannon
index was stable and low, around 1.94 ±0.15 (Figure 5). *Stainforthia fusiformis* dominated the
assemblage with relative abundances up to 56 % and BFAR up to 608 specimens.cm$^{-2}$.y$^{-1}$
(Figures 5A, 5B), which is the highest BFAR observed for this species along the core.
*Egerelloides medius/scabrus* was still very abundant, up to 48 % (Figure 5A). *Ammonia batava*,
the three *Elphidium* species and *N. labradorica* were present but with lower abundances than in
the zone FOR-A1 (maximum 5 %). *Bulimina marginata* started to be more abundant with an

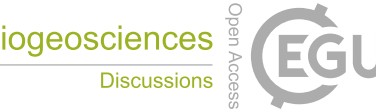

average relative abundance of 2 % in the zone. *Reophax subfusiformis* was still a part of the
assemblage and ranged between 1 and 8 %. The TOC and C/N values were stable and were on
average 3.5 % and 8.74 %, respectively (Figure 7). The clay size fraction dominated the
sediment during this period with a mean value of 63 %, and the sand content was around 6 %
(Figure 7).

423. Zone FOR-B1 (1953–1998)

The BFAR increased massively during the zone FOR-B1 with on average 54 ±31 specimens.cm$^{-}$
$^2$.y$^{-1}$ and with a peak at 93 specimens.cm$^{-2}$.y$^{-1}$ around 1965 (Figure 5). It is lower during the
second part of the zone. The Shannon index was higher than in previous zones and it
progressively increased towards the top of the zone (Shannon index average 2.34 ±0.3) (Figure
5). The highest BFAR along the core were observed for all the dominant species of the previous
zone FOR-A2, except for *S. fusiformis* (Figure 5B). The zone was then also characterized by a
drastic drop in the relative abundance of *S. fusiformis* from 31 to 2 % (Figure 5A).
*Eggerelloides medius/scabrus* gradually decreased in the zone, with relative abundances from
49 to 24 %. The highest relative abundance of *A. batava* for the entire record was in this zone but
it was slowly decreasing as well, from 10 to 3 %. The *Elphidium* group was more abundant than
in the FOR-A zones and their relative abundance was increasing, especially for *E. clavatum*
(increasing up to 23 %). *Bulimina marginata*, *N. labradorica* and *R. subfusiformis* had a relative
abundance between 2 and 6 %. A period of lower TOC values was observed during zone FOR-
B1 between 1953 and 1981, with an average of 2.38 % (Figure 7). On the same period, the sand
content showed a pronounced increase, with an average of 24 % (Figure 7).

424. Zone FOR-B2 (1998–2009)



In zone FOR-B2 the BFAR was still high, on average 55 ±6 specimens.cm$^{-2}$.y$^{-1}$ (Figure 5). The
Shannon index was high with an average of 2.8 ±0.2 (Figure 5). The dominant species in the
zone were *E. clavatum* (up to 25 %) and *Eggerelloides medius/scabrus* (up to 15 %; Figure 5A).
The other two *Elphidium* species reached their highest relative abundances over the core (up to
6 %). *Nonionella* sp. T1, which had not occurred in the record until now, appeared in this zone
with a relative abundance of 1 %. *Nonionoides turgida*, which was present in very low
abundances along the core, had a mean abundance of 1 % in the zone (Figure 6A). *Stainforthia*
*fusiformis* was present with up to 9 % in relative abundance and a BFAR higher than in zone
FOR-B1 (up to 570 specimens.cm$^{-2}$.y$^{-1}$). *Ammonia batava*, *B. marginata*, *N. labradorica*, and *R.*
*subfusiformis* were present and ranged between 2 and 8 %. The TOC values were increasing,
with on average 3.05 % (Figure 7). The sediment was dominated by the clay fraction that was
increasing (mean value of 58 %), and the sand content was around 17 % (Figure 7).
425. Zone FOR-C (2009–2013)
The BFAR was lower than in previous zones FOR-B1 and FOR-B2, with on average 21 ±5
specimens.cm$^{-2}$.y$^{-1}$ (Figure 5). The Shannon index was the highest during FOR-C (Shannon
index average 2.93 ±0.07) (Figure 5). *Nonionella* sp. T1 was a dominant specie in the zone with
a strong increase in relative abundance (from 1 to 14 %) and in BFAR (from 61 to 137
specimens.cm$^{-2}$.y$^{-1}$) (Figures 5A, 5B). *Elphidium clavatum* and *R. subfusiformis* were also
dominant species with abundances up to 13%. *Nonionoides turgida* had its highest relative
abundance and BFAR over the core during the zone, with up to 9 % and 342 specimens.cm$^{-2}$.y$^{-1}$,
respectively (Figures 5A, 5B). *Eggerelloides medius/scabrus* had its lowest relative abundance
over the core (up to 9 %). *Bulimina marginata*, the other two *Elphidium* species, *N. labradorica*
and *S. fusiformis* were still present (between 1 and 6 %), while *Ammonia batava* was absent





during the zone. The TOC and C/N values were on average 3.71 % and 8.17 %, respectively
(Figure 7). The clay size fraction dominated the sediment with a mean value of 66 % and the
sand fraction was 7 % (Figure 7).

426. Inner organic linings

Decalcified specimens were few and ranged between 0 and 4 specimens.cm$^{-2}$.y$^{-1}$ with an average
of 1 specimen.cm$^{-2}$.y$^{-1}$ (Fig. 5). They were observed throughout the core and especially during
zone FOR-B2, and the morphology of the remaining inner organic linings allowed the
identification of the taxon *Ammonia* (Plate 1).
4.3 Simulated data from model ECOSMO II
The VAV (vertically averaged South-North current velocity) through the Öresund from the
model ECOSMO II showed a reversed pattern compared to the UAV (vertically averaged West-
East current velocity) through the Kattegat (Figure 8). Thus, higher VAV through the Öresund
translates to an increase in the East to West flow in the Kattegat (lower UAV), suggesting a
stronger outflow from the Baltic Sea. The VAV through the Öresund had the lowest values
around 1955 (Figure 8), followed by a shift to very high values, which dominated throughout
1960–70. A comparable period with increased outflow from the Baltic into the Kattegat re-
occurred during the period 1993–2000.
5 – Discussion
Our environmental interpretations of the foraminiferal assemblages were based on the ecological
characteristics of each major species (Table 2). Based on our environmental reconstructions, we
could infer environmental changes regarding [O$_2$], salinity, organic matter content, and pollution



levels. Furthermore, we linked local environmental changes to larger atmospheric and
hydrographic conditions.
5.1. 1807 – 1870
All the major species found in this period are tolerant to low oxygen conditions, especially the
two main species: *S. fusiformis* and *E. medius/scabrus* (Table 2). *Stainforthia fusiformis* is an
opportunistic species used to hypoxic and potentially anoxic conditions (Alve 1994), and *E.*
*medius/scabrus* specimens have been found alive down to 10 cm in the sediment, where no
oxygen was available (Cesbron et al. 2016). *Stainforthia fusiformis* and *N. labradorica* are also
able to denitrify (Piña-Ochoa et al. 2010). The fact that species tolerant to low oxygen conditions
dominated, and the presence of species that have the capacity to denitrify, suggest that low
oxygen conditions were prevailing during this period. Furthermore, *S. fusiformis* prefers organic
rich substrate and clayey sediment, which was measured in our core during this time period
(Figure 7). The low species diversity, as indicated by the low Shannon index in this section of
the core, is usually linked with low salinity (Sen Gupta 1999a). Most of the major species found
during this period, such as the *Elphidium* group, *R. subfusiformis* and *A. batava* tolerate lower
salinities, and are typical of brackish environments (Table 2). The low occurrence of *B.*
*marginata*, a typical marine species, also suggests a salinity lower than in the open ocean.
However, the salinity was probably not below 30, which is the lower limit for *N. labradorica* and
*S. fusiformis*, which were present throughout the period (Figure 5, Table 2). In summary, this
period appears to have been characterized by low [$O_2$], high organic matter content, and salinity
around 30.
5.2 1870 – 1953



*Stainforthia fusiformis* was largely dominating the assemblage during this period, which may
suggest even lower oxygen conditions than during the previous period. This would also go along
with the low species diversity, which is usually linked to low salinity. However, the occurrence
of the marine species *B. marginata* suggests that the salinity was at least 32. Low oxygen is
frequently associated with high organic matter contents, since oxygen is consumed during
remineralization of organic matter. The TOC levels observed in our core in this zone were high,
but not higher than in the previous zone (Figure 7). At the time of the industrial revolution, the
Öresund was used as a sewage recipient for a mixture of domestic and industrial wastes,
industrial cooling water and drainage water (Henriksson 1968), and the amount of marine traffic
increased considerably during this time period. This diverse type of pollution could have
modified the water properties, for example regarding the carbonate chemistry and pH. Indeed,
this zone is characterized by the presence of organic linings in the core (see also section 5.6).
Moreover, heavy metals, fuel ash (black carbon) and pesticides have been demonstrated to
generally have a negative effect on foraminiferal abundance and diversity (Yanko et al. 1999).
Pollution and low oxygen concentration could explain the low species BFAR and diversity as
well as the dissolution of tests during this period. Other species that were present, i.e. the
agglutinated species *E. medius/scabrus* and *R. subfusiformis*, are known to be tolerant to various
kind of pollution (Table 2).
5.3 1953 – 1998
During this period, the large increase in the general BFAR could suggest that the specimens were
not in situ, but transported into the area. In line with this is the coarser grain size observed during
this period, indicating possible changes in the current system (Figure 7). However, the dating of
our core showed continuous sediment accumulation without any interruption during this period




(Figure 4). Moreover, all the new dominating species were already present in the core, even if in
lower relative abundances (Figure 5A). This indicates that the BFAR increase is most likely not
due to specimens transport but rather as a result of a change in substrate and environmental
conditions that became favorable for a different foraminiferal assemblage. The higher
foraminiferal diversity compared to previous periods and the decrease in the relative abundance
of *S. fusiformis* may indicate more oxic conditions. *Elphidium clavatum* has been found in coarse
sediment in the area (Bergsten et al. 1996), and other species that tolerate sandy environments
and varying TOC dominated the assemblage, such as *A. batava*, the other species in the
*Elphidium* group, *B. marginata*, and *E. medius/scabrus*. Furthermore, anthropogenic activities
such as agricultural practices were intensified during this period until the 1980s, which resulted
in increased nutrient loads and resulting eutrophication (i.e. Rydberg et al. 2006). The increase in
organic matter may have been beneficial for foraminifera as food source. Food webs and species
interaction like intra and inter competition might also have been modified, giving the advantage
to some species such as the *Elphidium* group to develop in these new environmental conditions.
The temporal coincidence with the shifts seen in the sediment record and the anomalous wind
conditions suggests a notable change of the currents through the Öresund (Figures 8, 9). The
simulated currents through the Öresund confirm such an abrupt change characterized by a shift
from very limited outflow from the Baltic to the Kattegat before ~1960 to more than a decade of
high relative outflow (high VAV) from the Öresund to the Kattegat and high current velocities
(Figure 8). While the simulation only covers the period after 1950, the analysis of wind
conditions and the NAO index suggest that the anomalies in the current and sediment pattern
from ~mid 1950's might have been unprecedented since at least the middle of the 19[th] century
(Figure 9). The shift in local sediment properties and the shift to higher BFAR and species



diversity suggest a combination of anomalous currents during a period of unusually negative
NAO index and the abrupt first advection of anthropogenic eutrophication from the Baltic Sea
towards the Kattegat. Consistent with our findings, long-term variations in Large Volume
Changes in the Baltic Sea (LVS, Lehmann and Post 2015; Lehmann et al. 2017), which are
calculated from >29 cm (~100 km³) daily sea-level changes at Landsort (58.74° N; 17.87° E) for
1887–2015, show an unusual cluster of both, more frequent and also larger LVCs during the
1970's to 1980's relative to the entire time period. Notably, this period coincides with the most
dramatic shift in foraminiferal BFAR and species diversity as well as an increase in sand content.
The period before the "regime shift" of the 1950's to 1960's is dominated by very infrequent and
few large LVC events. After the shift, the 1990's show also very few or partly no LVC events
with generally record-low Major Baltic Inflow events.
Thus, during this period, the ecosystems were affected both by climatic effects through
sedimentation changes, and human impact. After ~1980, the general BFAR was lower during a
short time (Figures 5, 9). This could be linked to the measures that were taken in agriculture and
water treatments in order to reduce the nutrients discharge (Carstensen et al. 2006; Conley et al.
2007), which could have reduced the food input. Interestingly, when the sedimentation pattern
changes again and the sand content decreases markedly (Figure 7), the new species in the
foraminiferal fauna do not return to previous relative abundances as one could have expected
(Figure 5A). This suggests that once the foraminiferal fauna was established in the Öresund area
after the ~1953 shift, it created a new state of equilibrium.
5.4 1998 – 2009
The foraminiferal assemblage in this zone was similar to the previous one, with high BFAR, high
diversity, and the *Elphidium* group as dominating species. This period is, however, characterized
by the appearance of two new major species: *N. turgida* and *Nonionella* sp. T1. *Nonionella* sp.
T1 is suggested to be an invasive species in the region which arrived by ship ballast tanks around
1985, and rapidly expanded to the Kattegat and Öresund (Polovodova Asteman and Schönfeld
2015). According to our dated core, the species arrived in the Öresund ~2000 CE (Figure 5). The
species is also present on the south coast of Norway since after 2009 (Deldicq et al., in press),
but additional genetic analyses are necessary to have a better overview of the species' origin and
expansion. *Nonionoides turgida* is an opportunistic species that prefers high levels of organic
matter in the sediment, as observed in our core during this period (Figure 7). The increase in the
*S. fusiformis* BFAR suggest lower [$O_2$] than in the previous zone, which was indeed a general
trend in the Danish waters during this time period (Conley et al. 2007). This period was then
characterized by low [$O_2$], high organic matter content, and open ocean salinity.
5.5 2009 – 2013
The ability of *Nonionella* sp. T1 to denitrify and its tolerance to varying environment may
explain its rapid increase during this period. The increase of *N. turgida* also suggests higher
levels of organic matter in the sediment. The dominance of these two species and the lower
BFAR compared to previous periods suggest low oxygen levels. This period is thus characterized
by low [$O_2$], high organic matter content, and open ocean salinity.
5.6 Dissolution
The inner organic linings of the taxon *Ammonia* were observed (in low numbers, < 5 units) along
the whole core, except in the top two centimeters (Figure 5). Inner organic linings of the taxa



*Ammonia* and/or *Elphidium* were noticed in previous studies among dead fauna in the region
(Jarke 1961; Hermelin 1987: Baltic Sea; Christiansen et al. 1996; Murray and Alve 1999:
Kattegat and Skagerrak; Filipsson and Nordberg 2004b: Koljö Fjord). Dissolution of calcareous
foraminiferal tests has been considered as a taphonomic process, affecting the test of the
specimens after their death (Martin 1999; Berkeley et al. 2007). However, living decalcified
foraminifera have been observed in their natural environment in the south Baltic Sea (Charrieau
et al. 2018) and the Arcachon Bay, France (Cesbron et al. 2016) and, proving that test dissolution
can also occur while the specimens live. In any case, low pH and low calcium carbonate
saturation are suggested as involved in the observed dissolution (Jarke 1961; Christiansen et al.
1996; Murray and Alve 1999; Cesbron et al. 2016; Charrieau et al. 2018). Test dissolution may
occur in all calcitic species, but only the organic linings of *Ammonia* were found in our study,
probably because these were more robust to physical stress such as abrasion.
6 – Conclusion
In this study, we described an environmental record from the Öresund, based on benthic
foraminifera – and geochemical data and we link the results with reconstructed wind data, NAO
index and current changes model. Five foraminiferal zones were differentiated and associated
with environmental changes in terms of salinity, $[O_2]$, and organic matter content. The main
event is a major shift in the foraminiferal assemblage ~1950, when the BFAR massively
increased and *S. fusiformis* stopped dominating the assemblage. This period also corresponds to
an increase in grain-size, resulting in a higher sand content. The grain-size distribution suggests
changes in the current velocities which are confirmed by simulated current velocity through the
Öresund. Human activities through increased eutrophication also influenced the foraminiferal





fauna changes during this period. Organic linings of *Ammonia* were observed throughout the
core, probably linked to low pH and calcium carbonate saturation, affecting test preservation.
The long-term reconstruction of sediment – and ecosystem parameters since ~1807 suggests that
the onset of increased anthropogenic eutrophication of the eastern Kattegat started with an abrupt
shift ~1960 during a period of strongly negative NAO. With unusually calm wind conditions
during the winter half and increased easterly winds, the conditions were ideal for larger Baltic
outflow invents which then allowed more frequent and larger Baltic inflow events, as calculated
from LVC events during this period. The sediment record with unprecedented high temporal
resolution points towards the importance of considering also large Baltic outflow events to the
Kattegat which have a large impact at least at Ven Island and possibly larger parts of the
Kattegat. Because the Baltic Sea has much higher eutrophication levels and less oxygenated and
less saline waters, larger outflow events may have a significant impact also on the Kattegat.
Periods with negative NAO or conditions with intense atmospheric blocking over Scandinavia
like 2018 may also increase the influence of Baltic Sea problems on the Kattegat region.
Acknowledgments
We would like to thank the captain and the crew of the r/v *Skagerak*. We acknowledge Git
Klintvik Ahlberg for the assistance in the laboratory, Yasmin Bokhari Friberg and Åsa Wallin
for the help with the grain-size analysis, and Guillaume Fontorbe for help with the age model.
The hydrographic data used in the projected is collected from SMHI's data base SHARK. The
SHARK data collection is organized by the environmental monitoring program and funded by
the Swedish Environmental Protection Agency. The study was financially supported by the





Swedish Research Council FORMAS (grants 2012-2140 and 217-2010-126), the Royal
Physiographic Society in Lund and Oscar and Lili Lamm's Foundation.
Supplementary data
Appendix A with time series of salinity, temperature and dissolved oxygen concentration at the
bottom water of the Öresund and Appendix B with total foraminiferal faunas normalized to 50
cm$^3$ along the DV core are available in the online version of the article.

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

Figures
Figure 1. Map of the studied area. The star shows the focused station of this study. General water
circulation: main surface currents (black arrows) and main deep currents (grey arrows). GB:
Great Belt; LB: Little Belt; AW: Atlantic Water; CNSW: Central North Sea Water; JCW; Jutland
Coastal Water; NCC: Norwegian Coastal Current; BW: Baltic Water. Insert source: © BSHC.
Figure 2. CTD profiles of temperature, salinity, pH and dissolved oxygen concentration in the
water column for the DV-1 station (modified from Charrieau et al. 2018).
Figure 3. Seasonal variability of salinity, temperature, pH and dissolved inorganic nitrogen
(DIN) concentration at the surface water (light grey), and seasonal variability of salinity,
temperature, pH and dissolved oxygen concentration at the bottom water (40-50 m) (dark grey)
of the Öresund. The data were measured between 1965 and 2016 by the SMHI (Swedish





Meteorological and Hydrological Institute) at the station W LANDSKRONA. The number of
measurements is indicated for each month.
Figure 4. Age-depth calibration for the sediment sequence from the Öresund (DV-1). A) Total
and supported [210]Pb activity. B) Unsupported [210]Pb activity and the associated age-model. C)
[137]Cs activity. The peak corresponds to the Chernobyl reactor accident in 1986. D) Age-depth
model for the whole sediment sequence based on [210]Pb dates and calculated sediment
accumulation rates (SAR).
Figure 5. A) Relative abundances (%) of the foraminiferal major species (>5 %), benthic
foraminiferal accumulation rate (BFAR, specimens.cm$^{-2}$.yr$^{-1}$), Shannon index, organic linings
(specimens.cm$^{-2}$.yr$^{-1}$) and factors from the correspondence analysis. B) Benthic foraminiferal
accumulation rates (specimens.cm$^{-2}$.yr$^{-1}$) of the major species (>5%), BFAR (specimens.cm$^{-2}$.yr$^{-}$
$^{1}$), Shannon index, organic linings (specimens.cm$^{-2}$.yr$^{-1}$) and factors from the correspondence
analysis. Foraminiferal zones based on cluster analysis. Note the different scale on the x axes.
Figure 6. Dendrogram produced by the cluster analysis based on the Morisita index and the
UPGMA clustering method.
Figure 7. Sediment parameters of the cores DV-1I and DV-1G ([210]Pb dated): total organic carbon
content (C$_{org}$) (%), C/N ratio, and grain size (%). Foraminiferal zones indicated.
Figure 8. South-North flow (VAV) in the Öresund (dark line) and West-East flow (UAV) in the
Kattegat (light line) between 1950 and 2013. Foraminiferal zones indicated.
Figure 9. A) NAO index for boreal winter (December to March), data from Jones et al. (1997).
B) Variations of near-surface (10 m) wind conditions (October to March), data from Schenk and



Zorita (2012). Both NAO index and wind speed data are normalized on the period 1850-2008
and show running decadal means. C) BFAR, percentage of sand fraction and West-East flow
(UAV) in the Kattegat. Foraminiferal zones indicated.
Plate 1. SEM pictures of the major foraminiferal species (>5%). 1. *Stainforthia fusiformis*; 2.
*Nonionellina labradorica*; 3. *Nonionella* sp. T1; 4. *Nonionoides turgida*; 5. *Eggerelloides*
*medius/scabrus*; 6. *Bulimina marginata*; 7. *Ammonia batava*; 8. *Reophax subfusiformis*; 9.
*Elphidium magellanicum*; 10. *Elphidium clavatum*; 11-12. *Ammonia* sp.
Tables
Table 1. Significant foraminiferal species and scores according to the correspondence analysis.
Table 2. Ecological significance of the benthic foraminiferal assemblages (major species).
Appendix
Appendix A. Time series of salinity, temperature and dissolved oxygen concentration at the
bottom water (40 m) of the Öresund between 1986 and 2013. The data were measured by the
SMHI (Swedish Meteorological and Hydrological Institute) at the station W LANDSKRONA.
Appendix B. Total faunas, normalized to 50 cm³


















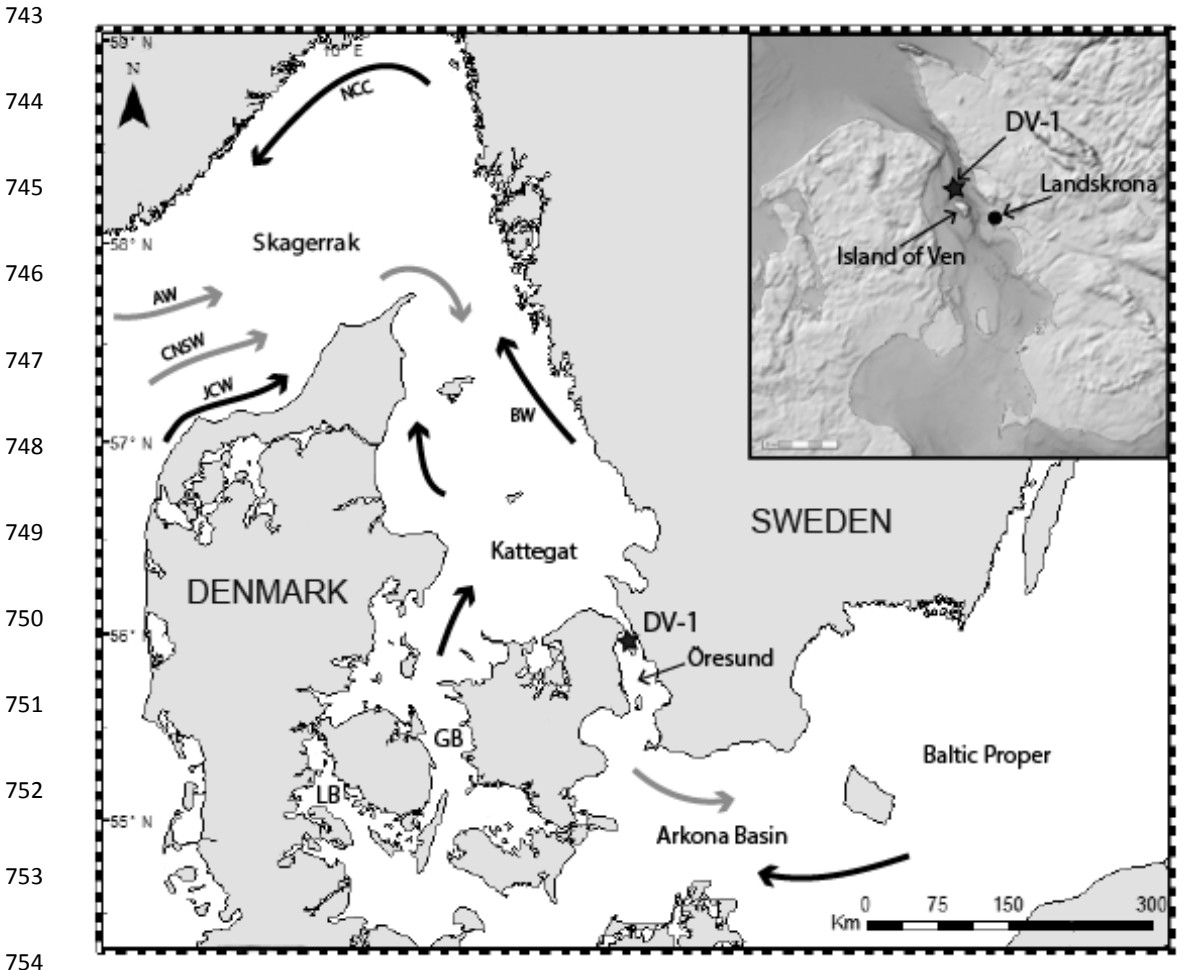

Figure 1. Map of the studied area. The star shows the focused station of this study. General water
circulation: main surface currents (black arrows) and main deep currents (grey arrows). GB:
Great Belt; LB: Little Belt; AW: Atlantic Water; CNSW: Central North Sea Water; JCW; Jutland
Coastal Water; NCC: Norwegian Coastal Current; BW: Baltic Water. Insert source: © BSHC.





















Figure 2. CTD profiles of temperature, salinity, pH and dissolved oxygen concentration in the
water column for the DV-1 station (modified from Charrieau et al. 2018).
















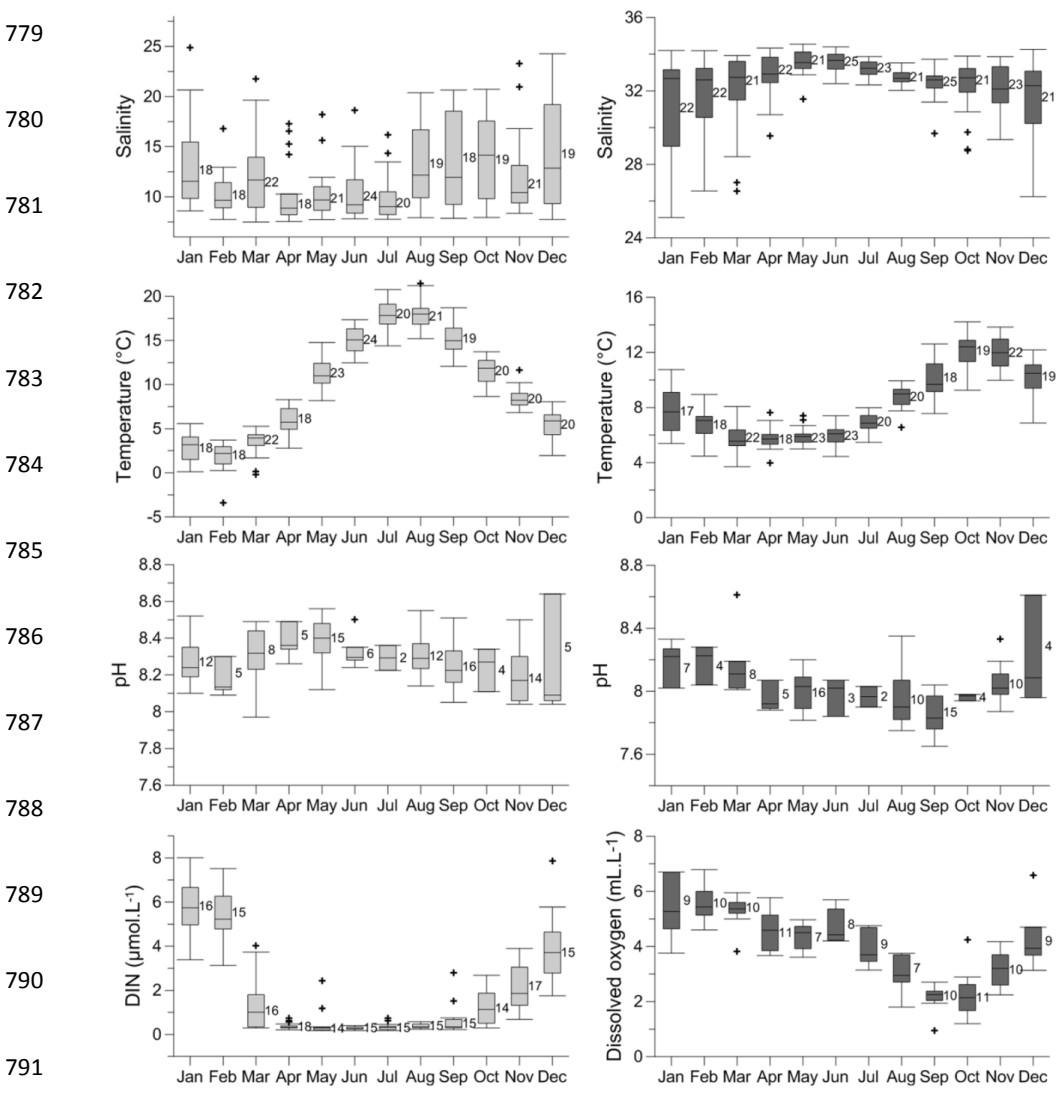

Figure 3. Seasonal variability of salinity, temperature, pH and dissolved inorganic nitrogen
(DIN) concentration at the surface water (light grey), and seasonal variability of salinity,
temperature, pH and dissolved oxygen concentration at the bottom water (40-50 m) (dark grey)
of the Öresund. The data were measured between 1965 and 2016 by the SMHI (Swedish
Meteorological and Hydrological Institute) at the station W LANDSKRONA. The number of
measurements is indicated for each month.



Figure 4. Age-depth calibration for the sediment sequence from the Öresund (DV-1). A) Total and supported [210]Pb activity. B) Unsupported [210]Pb activity and the associated age-model. C) [137]Cs activity. The peak corresponds to the Chernobyl reactor accident in 1986. D) Age-depth model for the whole sediment sequence based on [210]Pb dates and calculated sediment accumulation rates (SAR).



Figure 5. A) Relative abundances (%) of the foraminiferal major species (>5 %), benthic foraminiferal accumulation rate (BFAR, specimens.cm$^{-2}$.yr$^{-1}$), Shannon index, organic linings (specimens.cm$^{-2}$.yr$^{-1}$) and factors from the correspondence analysis. B) Benthic foraminiferal accumulation rates (specimens.cm$^{-2}$.yr$^{-1}$) of the major species (>5%), BFAR (specimens.cm$^{-2}$.yr$^{-1}$), Shannon index, organic linings (specimens.cm$^{-2}$.yr$^{-1}$) and factors from the correspondence analysis. Foraminiferal zones based on cluster analysis. Note the different scale on the x axes.





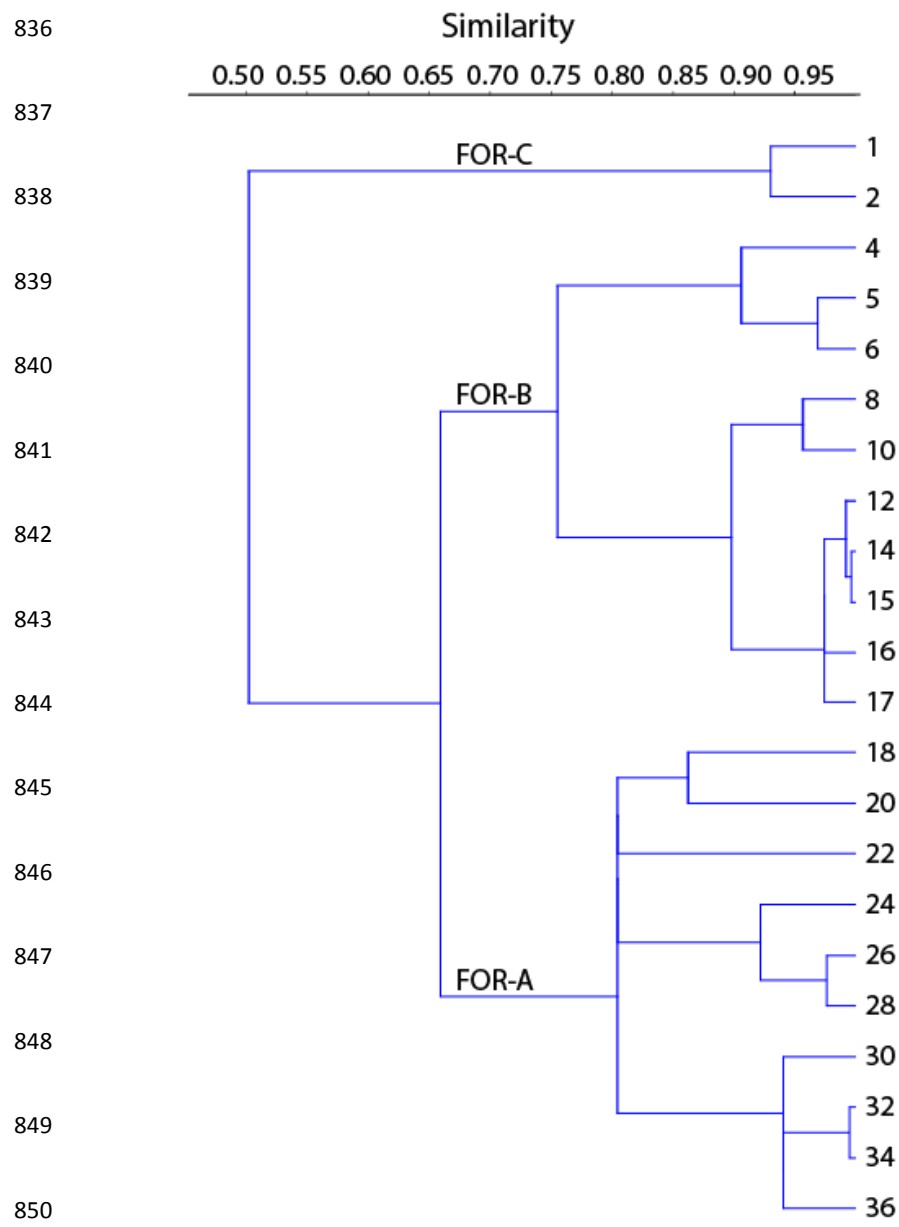

Figure 6. Dendrogram produced by the cluster analysis based on the Morisita index and the UPGMA clustering method.



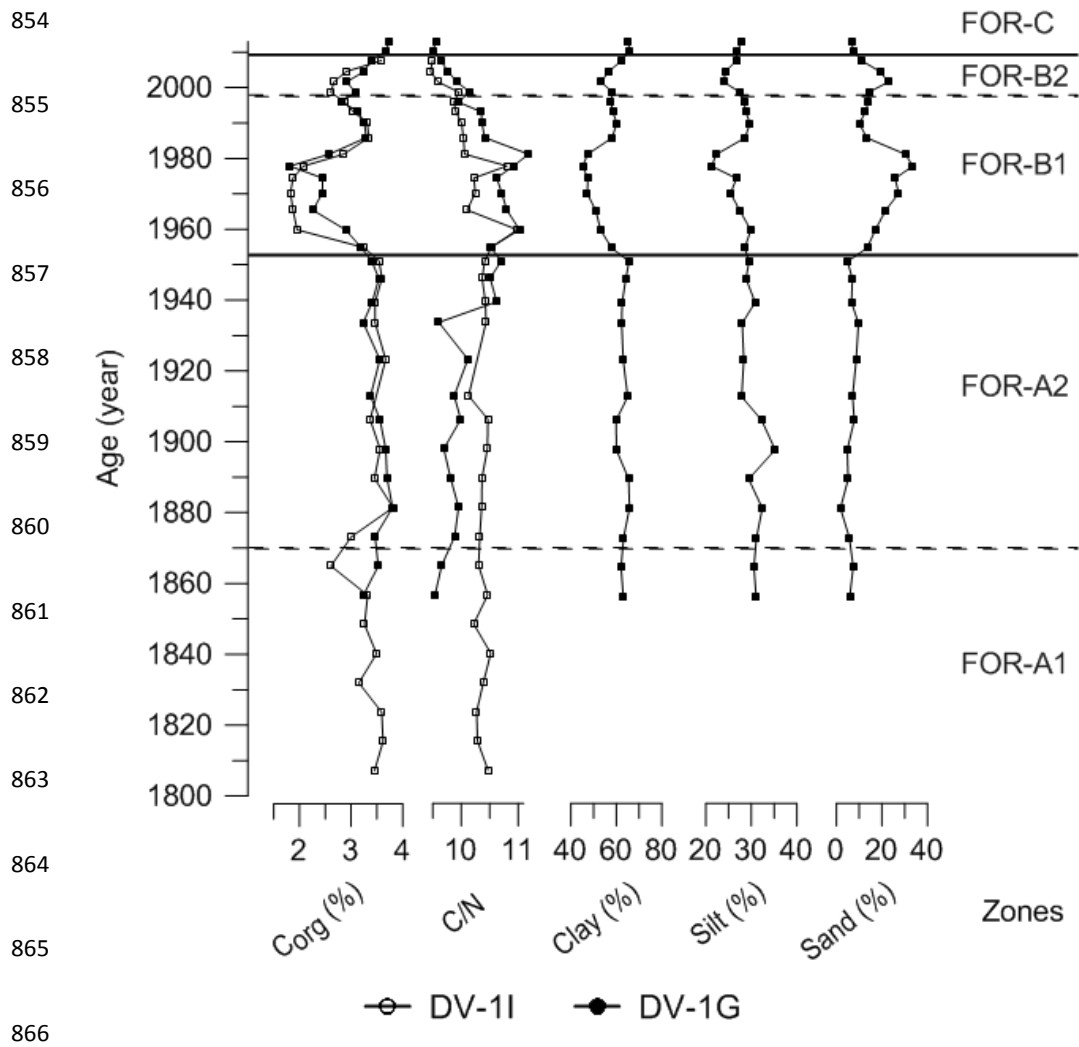

Figure 7. Sediment parameters of the cores DV-1I and DV-1G ($^{210}$Pb dated): total organic carbon

content ($C_{org}$) (%), C/N ratio, and grain size (%). Foraminiferal zones indicated.













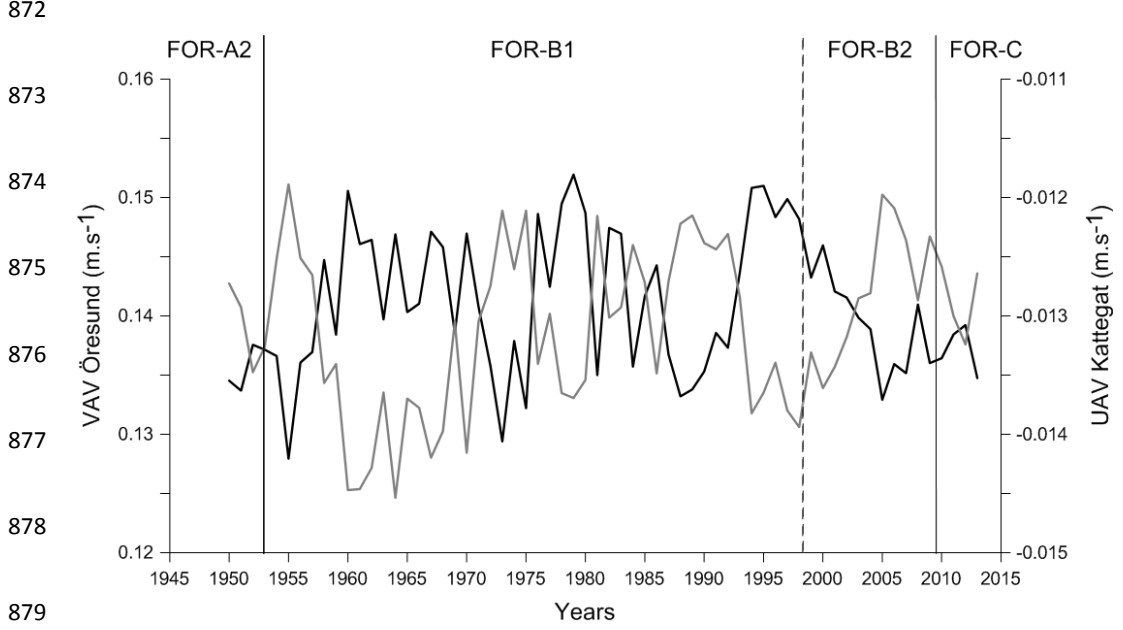

Figure 8. South-North flow (VAV) in the Öresund (dark line) and West-East flow (UAV) in the
Kattegat (light line) between 1950 and 2013. Foraminiferal zones indicated.








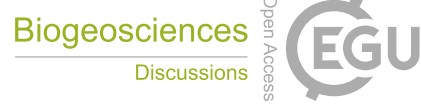

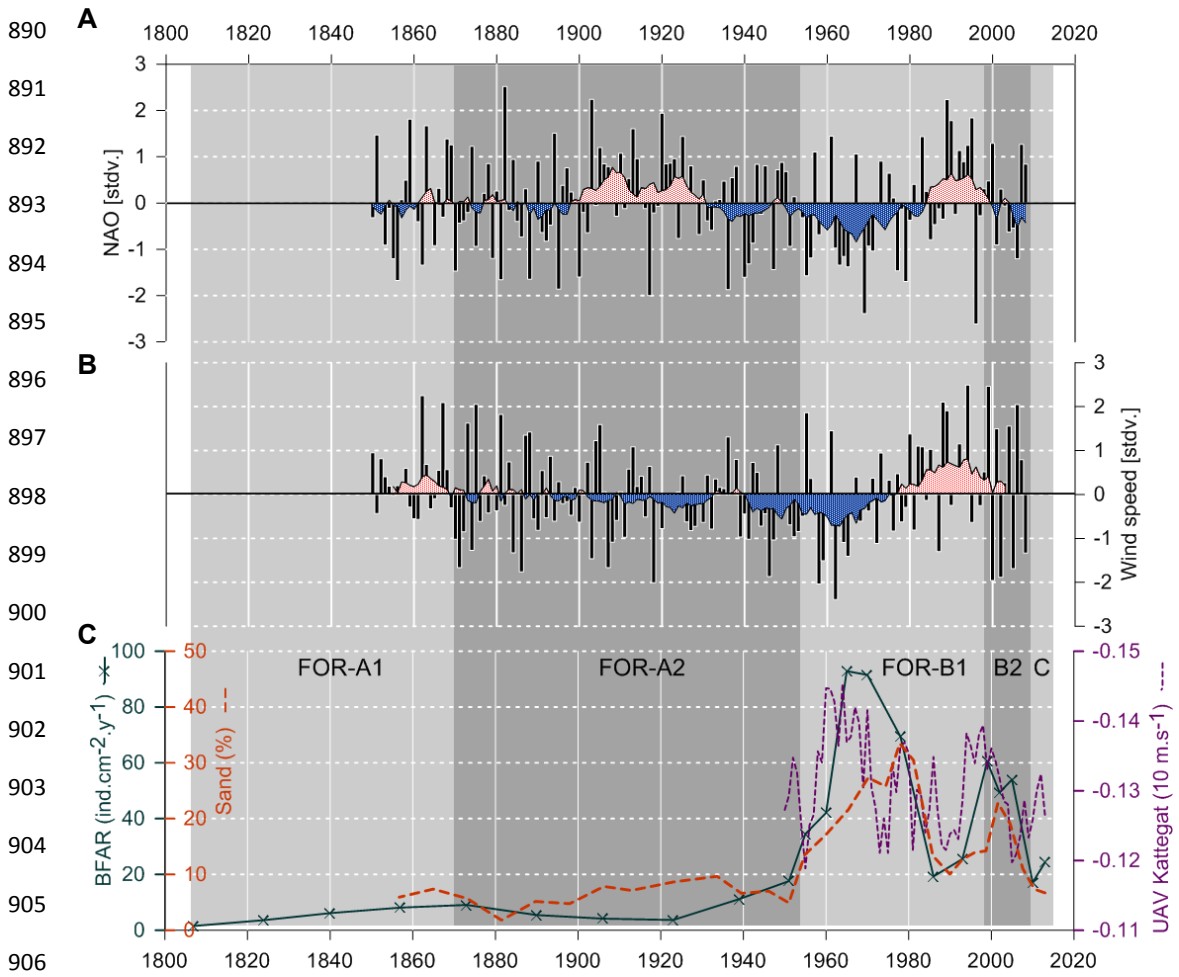

Figure 9. A) NAO index for boreal winter (December to March), data from Jones et al. (1997).

B) Variations of near-surface (10 m) wind conditions (October to March), data from Schenk and

Zorita (2012). Both NAO index and wind speed data are normalized on the period 1850-2008

and show running decadal means. C) BFAR, percentage of sand fraction and West-East flow

(UAV) in the Kattegat. Foraminiferal zones indicated.






Plate 1. SEM pictures of the major foraminiferal species (>5%). 1. *Stainforthia fusiformis*; 2.

*Nonionellina labradorica*; 3. *Nonionella* sp. T1; 4. *Nonionoides turgida*; 5. *Eggerelloides*

*medius/scabrus*; 6. *Bulimina marginata*; 7. *Ammonia batava*; 8. *Reophax subfusiformis*; 9.

*Elphidium magellanicum*; 10. *Elphidium clavatum*; 11-12. *Ammonia* sp.






Table 1. Significant foraminiferal species and scores according to the correspondence analysis.

| Factor | Total variance (%) | Significant species | Score |
|--------|--------------------|---------------------|-------|
| 1 | 48.18 | *Nonionella* sp. T1 | 5.10 |
| | | *Nonionoides turgida* | 4.14 |
| 2 | 30.88 | *Ammonia batava* | 1.34 |
| | | *Stainforthia fusiformis* | -1.41 |
| 3 | 13.36 | *Elphidium albiumbilicatum* | -1.65 |
| | | *Elphidium clavatum* | -1.57 |
| | | *Elphidium magellanicum* | -1.32 |

















Table 2. Ecological significance of the benthic foraminiferal assemblages (major species).

| Species | Ecological significance | Reference |
|---|---|---|
| *Ammonia batava* | Salinity 15-35, T 0-29$^0$C, high tolerance to varying substrate and TOC | Alve and Murray (1999); Murray (2006) |
| *Bulimina marginata* | Tolerates low oxygen conditions, salinity 30-35, T 5-13$^0$C, muddy sand, prefers organic rich substrates | Conradsen (1993); Murray (2006) |
| *Elphidium albiumbilicatum* | Salinity 16-26, typical brackish species | Alve and Murray (1999) |
| *Elphidium clavatum* | Tolerates low oxygen conditions, salinity 10-35, T 0-7$^0$C, high tolerance to varying substrate and TOC, subtidal | Conradsen {Citation}(1993); Alve and Murray (1999); Murray (2006) |
| *Elphidium magellanicum* | Coastal species | Sen Gupta (1999) |
| *Nonionella stella*/aff. *stella* | Tolerates low oxygen conditions, kleptoplastidy, able of denitrification, invasive in the Skagerrak-Kattegat | Piña-Ochoa et al. (2010); Bernhard et al. (2012); Charrieau et al. (2018) |
| *Nonionellina labradorica* | Salinity >30, T 4-14$^0$C, high latitudes, kleptoplastidy, able of denitrification | Cedhagen (1991) |
| *Nonionoides turgida* | Opportunistic species, tolerates low oxygen conditions, prefers high food availability | Van der Zwaan and Jorissen (1991) |
| *Stainforthia fusiformis* | Opportunistic species, tolerates very low oxygen conditions, salinity >30, able of denitrification, prefers organic rich substrates, fast reproduction cycle | Alve (1994); Filipsson and Nordberg (2004); Piña-Ochoa et al. (2010) |
| *Eggerelloides medius/scabrus* | High tolerance to hypoxia, salinity 20-35, T 8-14$^0$C, sandy-muddy sand, tolerance to various kind of pollution | Alve and Murray (1999); Alve (1990); Murray (2006); Cesbron et al. (2016) |
| *Reophax subfusiformis* | Tolerance to environmental variations | Sen Gupta (1999) |






























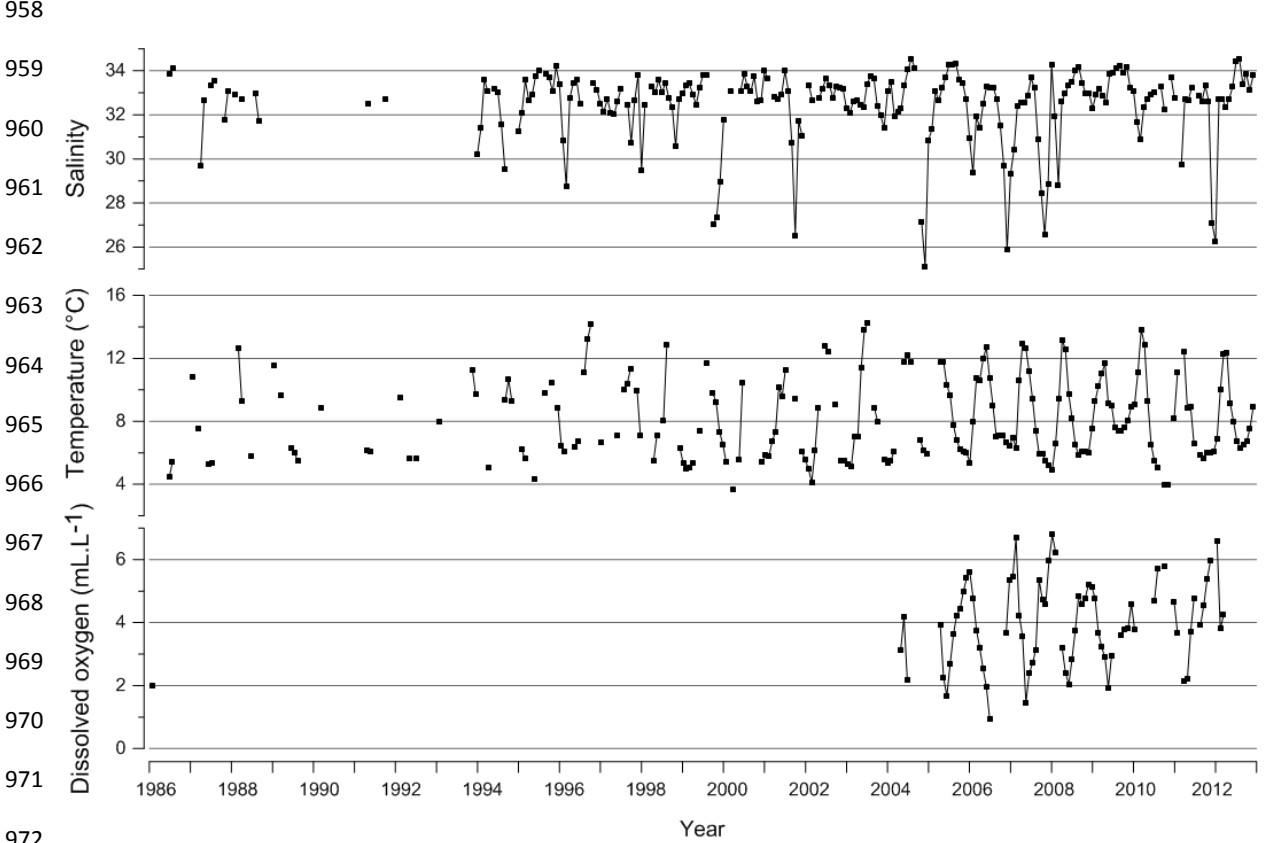

Appendix A. Time series of salinity, temperature and dissolved oxygen concentration at the
bottom water (40 m) of the Öresund between 1986 and 2013. The data were measured by the
SMHI (Swedish Meteorological and Hydrological Institute) at the station W LANDSKRONA.









Appendix B. Total faunas, normalized to 50 cm³

| Station name | | | | | | | | | DV | | | | | | | | | | | | | |
|---|---|---|---|---|---|---|---|---|---|---|---|---|---|---|---|---|---|---|---|---|---|---|
| FOR zones | FOR-C | | FOR-B2 | | | FOR-B1 | | | | | | | FOR-A2 | | | | | | FOR-A1 | | | |
| Centimeter | 1 | 2 | 4 | 5 | 6 | 8 | 10 | 12 | 14 | 15 | 16 | 17 | 18 | 20 | 22 | 24 | 26 | 28 | 30 | 32 | 34 | 36 |
| Species | | | | | | | | | | | | | | | | | | | | | | |
| *Biloculinella inflata* | 6 | 13 | 0 | 0 | 0 | 0 | 0 | 0 | 0 | 0 | 0 | 0 | 0 | 0 | 0 | 0 | 0 | 0 | 0 | 0 | 0 | 0 |
| *Cornuspira involvens* | 0 | 6 | 9 | 0 | 0 | 0 | 0 | 0 | 0 | 0 | 0 | 0 | 0 | 0 | 0 | 0 | 0 | 7 | 0 | 0 | 0 | 0 |
| *Pyrgo williamsoni* | 1 | 1 | 0 | 16 | 0 | 0 | 0 | 0 | 0 | 0 | 0 | 0 | 0 | 0 | 0 | 0 | 0 | 28 | 0 | 0 | 0 | 0 |
| *Quinqueloculina seminula* | 0 | 1 | 0 | 0 | 35 | 0 | 14 | 0 | 0 | 0 | 30 | 0 | 0 | 0 | 0 | 0 | 0 | 0 | 0 | 0 | 0 | 0 |
| *Quinqueloculina stalkeri* | 6 | 0 | 9 | 0 | 18 | 0 | 0 | 0 | 0 | 0 | 0 | 0 | 0 | 0 | 0 | 0 | 7 | 14 | 0 | 0 | 0 | 0 |
| Porcelaneous varia | 13 | 31 | 111 | 32 | 0 | 0 | 0 | 0 | 34 | 8 | 46 | 0 | 0 | 8 | 0 | 0 | 0 | 0 | 7 | 9 | 0 | 0 |
| Organic linings | 0 | 0 | 146 | 159 | 158 | 60 | 345 | 132 | 171 | 238 | 304 | 332 | 686 | 575 | 599 | 807 | 444 | 260 | 608 | 316 | 649 | 159 |
| *Ammonia beccarii* | 0 | 0 | 292 | 191 | 308 | 105 | 159 | 822 | 1495 | 2167 | 1033 | 498 | 123 | 121 | 15 | 16 | 14 | 49 | 57 | 103 | 56 | 25 |
| *Ammonia falsobeccarii* | 57 | 77 | 69 | 80 | 35 | 37 | 111 | 350 | 854 | 986 | 516 | 231 | 85 | 23 | 15 | 0 | 0 | 0 | 0 | 0 | 0 | 0 |
| *Ammonia* spp. | 0 | 0 | 0 | 0 | 0 | 142 | 0 | 0 | 0 | 0 | 0 | 0 | 0 | 0 | 0 | 0 | 0 | 0 | 0 | 0 | 0 | 0 |
| *Bolivina pseudoplicata* | 0 | 0 | 9 | 0 | 0 | 0 | 0 | 0 | 68 | 0 | 0 | 0 | 0 | 0 | 0 | 0 | 0 | 0 | 0 | 0 | 0 | 2 |
| *Bolivina pseudopunctata* | 19 | 0 | 0 | 0 | 0 | 0 | 0 | 0 | 0 | 0 | 0 | 0 | 0 | 0 | 0 | 0 | 0 | 0 | 0 | 0 | 0 | 0 |
| *Bolivina* spp. | 0 | 0 | 9 | 0 | 0 | 0 | 0 | 19 | 0 | 0 | 0 | 0 | 0 | 0 | 0 | 0 | 0 | 0 | 0 | 0 | 0 | 0 |
| *Bulimina marginata* | 132 | 107 | 506 | 414 | 282 | 187 | 166 | 661 | 1128 | 1224 | 501 | 534 | 116 | 68 | 29 | 16 | 57 | 7 | 0 | 0 | 8 | 6 |
| *Buliminella elegantissima* | 0 | 6 | 206 | 143 | 176 | 60 | 83 | 57 | 103 | 170 | 61 | 29 | 8 | 8 | 7 | 8 | 0 | 70 | 7 | 9 | 0 | 0 |
| *Cassidulina laevigata* | 44 | 101 | 300 | 112 | 35 | 22 | 0 | 340 | 376 | 510 | 228 | 116 | 15 | 8 | 7 | 0 | 0 | 0 | 0 | 0 | 0 | 2 |
| *Cassidulina reniforme* | 0 | 13 | 17 | 32 | 0 | 15 | 14 | 19 | 68 | 0 | 15 | 14 | 0 | 0 | 0 | 0 | 0 | 0 | 0 | 0 | 0 | 0 |
| *Cibicides lobatulus* | 63 | 57 | 352 | 287 | 211 | 22 | 41 | 359 | 410 | 238 | 273 | 130 | 8 | 8 | 0 | 16 | 0 | 7 | 14 | 43 | 8 | 8 |
| *Elphidium albiumbilicatum* | 25 | 63 | 489 | 143 | 528 | 225 | 180 | 454 | 410 | 238 | 213 | 217 | 77 | 53 | 15 | 31 | 29 | 127 | 78 | 77 | 0 | 14 |
| *Elphidium clavatum* | 201 | 289 | 986 | 1833 | 2077 | 809 | 567 | 1436 | 2631 | 3331 | 1018 | 1430 | 154 | 136 | 51 | 39 | 100 | 183 | 155 | 111 | 72 | 45 |
| *Elphidium magellanicum* | 63 | 94 | 292 | 223 | 528 | 135 | 180 | 529 | 547 | 408 | 349 | 130 | 62 | 45 | 0 | 0 | 43 | 141 | 92 | 60 | 8 | 8 |
| *Elphidium williamsoni* | 19 | 19 | 86 | 32 | 18 | 22 | 14 | 113 | 68 | 136 | 61 | 14 | 0 | 0 | 0 | 0 | 7 | 28 | 21 | 51 | 16 | 6 |
| *Elphidium* spp. | 69 | 126 | 86 | 0 | 53 | 0 | 28 | 0 | 0 | 0 | 0 | 0 | 0 | 15 | 0 | 0 | 21 | 7 | 14 | 17 | 8 | 2 |
| *Epistominella vitrea* | 19 | 13 | 309 | 367 | 299 | 120 | 166 | 227 | 103 | 204 | 30 | 43 | 23 | 0 | 7 | 0 | 7 | 0 | 0 | 0 | 0 | 0 |
| *Fissurina* spp. | 0 | 0 | 0 | 0 | 0 | 0 | 0 | 0 | 34 | 0 | 15 | 0 | 0 | 0 | 0 | 0 | 0 | 0 | 0 | 0 | 0 | 0 |
| *Parafissurina* spp. | 0 | 0 | 43 | 16 | 35 | 22 | 0 | 38 | 34 | 68 | 15 | 14 | 8 | 0 | 0 | 0 | 0 | 7 | 0 | 0 | 0 | 0 |
| *Fursenkoïna* spp. | 0 | 0 | 0 | 0 | 0 | 0 | 0 | 0 | 0 | 0 | 0 | 0 | 0 | 0 | 0 | 0 | 0 | 7 | 0 | 0 | 0 | 0 |
| *Gavelinopsis praegeri* | 0 | 6 | 0 | 0 | 0 | 0 | 0 | 0 | 0 | 0 | 15 | 0 | 0 | 0 | 0 | 0 | 0 | 0 | 0 | 0 | 0 | 0 |
| *Giroidina* sp. | 0 | 0 | 0 | 0 | 0 | 0 | 0 | 0 | 34 | 0 | 0 | 0 | 0 | 0 | 0 | 0 | 0 | 0 | 0 | 0 | 0 | 0 |
| *Haynesina depressula* | 25 | 25 | 51 | 0 | 0 | 0 | 0 | 0 | 0 | 0 | 0 | 0 | 0 | 0 | 0 | 0 | 0 | 0 | 0 | 17 | 0 | 0 |
| *Hyalinea balthica* | 0 | 19 | 9 | 0 | 0 | 7 | 0 | 0 | 34 | 0 | 0 | 0 | 0 | 0 | 0 | 0 | 0 | 0 | 0 | 0 | 0 | 0 |
| *Lagena laevis* | 0 | 0 | 0 | 0 | 0 | 7 | 0 | 0 | 0 | 0 | 15 | 0 | 0 | 0 | 7 | 0 | 7 | 14 | 0 | 0 | 0 | 0 |



Appendix B. Total faunas, normalized to 50 cm³

| Species | | | | | | | | | | | | | | | | | | | | | | |
|---|---|---|---|---|---|---|---|---|---|---|---|---|---|---|---|---|---|---|---|---|---|---|
| *Lagena semistriata* | 0 | 0 | 0 | 0 | 18 | 0 | 0 | 0 | 0 | 0 | 0 | 0 | 0 | 8 | 7 | 0 | 0 | 0 | 0 | 0 | 0 | 0 |
| *Lagena substriata* | 0 | 13 | 9 | 32 | 53 | 22 | 14 | 0 | 34 | 34 | 0 | 0 | 15 | 15 | 0 | 8 | 14 | 14 | 0 | 9 | 0 | 0 |
| *Lagena sulcata* | 0 | 0 | 9 | 0 | 18 | 0 | 0 | 0 | 103 | 34 | 0 | 0 | 8 | 0 | 0 | 0 | 0 | 0 | 0 | 9 | 0 | 0 |
| *Lenticulina* sp. | 0 | 0 | 0 | 48 | 0 | 7 | 0 | 0 | 0 | 34 | 0 | 0 | 0 | 0 | 7 | 0 | 0 | 0 | 0 | 9 | 8 | 0 |
| *Loxostomum* sp. | 0 | 0 | 9 | 0 | 0 | 0 | 0 | 57 | 0 | 0 | 0 | 0 | 0 | 0 | 0 | 0 | 0 | 7 | 0 | 0 | 0 | 2 |
| *Nonionella* sp. T1 | 308 | 176 | 94 | 0 | 18 | 0 | 0 | 0 | 0 | 0 | 0 | 0 | 0 | 0 | 0 | 0 | 0 | 0 | 0 | 0 | 0 | 0 |
| *Nonionella iridea* | 0 | 0 | 0 | 16 | 18 | 22 | 0 | 38 | 0 | 0 | 0 | 0 | 0 | 0 | 0 | 0 | 0 | 21 | 0 | 0 | 0 | 0 |
| *Nonionellina labradorica* | 113 | 75 | 249 | 143 | 141 | 135 | 97 | 340 | 513 | 382 | 243 | 188 | 54 | 23 | 22 | 16 | 29 | 56 | 106 | 103 | 40 | 12 |
| *Nonionoides turgida* | 138 | 189 | 103 | 64 | 106 | 0 | 0 | 38 | 34 | 34 | 15 | 0 | 0 | 0 | 0 | 0 | 0 | 7 | 0 | 0 | 0 | 2 |
| *Nonionella* spp. | 0 | 0 | 0 | 16 | 35 | 0 | 0 | 19 | 0 | 0 | 0 | 0 | 8 | 0 | 0 | 0 | 0 | 7 | 0 | 0 | 0 | 0 |
| *Nonionellina* spp. | 0 | 0 | 17 | 0 | 0 | 0 | 0 | 19 | 34 | 0 | 0 | 0 | 0 | 0 | 0 | 0 | 0 | 0 | 0 | 0 | 0 | 2 |
| *Oolina melo* | 6 | 0 | 0 | 0 | 0 | 0 | 0 | 19 | 0 | 68 | 0 | 0 | 0 | 0 | 0 | 0 | 0 | 0 | 0 | 0 | 0 | 0 |
| *Polymorphina* spp. | 0 | 0 | 9 | 16 | 0 | 0 | 0 | 38 | 0 | 0 | 15 | 0 | 0 | 15 | 0 | 0 | 7 | 0 | 7 | 9 | 0 | 0 |
| *Procerolagena clavata* | 0 | 6 | 0 | 0 | 0 | 0 | 0 | 0 | 0 | 0 | 0 | 0 | 0 | 0 | 0 | 0 | 0 | 0 | 0 | 0 | 0 | 0 |
| *Procerolagena grassilima* | 0 | 0 | 43 | 0 | 18 | 15 | 0 | 0 | 0 | 0 | 30 | 14 | 8 | 23 | 7 | 0 | 0 | 0 | 0 | 9 | 0 | 0 |
| *Procerolagena mollis* | 0 | 0 | 17 | 0 | 0 | 0 | 0 | 0 | 0 | 0 | 0 | 0 | 8 | 8 | 0 | 0 | 0 | 0 | 7 | 0 | 0 | 0 |
| *Robertina arctica* | 0 | 6 | 0 | 0 | 0 | 0 | 0 | 0 | 0 | 0 | 0 | 0 | 0 | 0 | 0 | 0 | 0 | 0 | 0 | 0 | 0 | 0 |
| *Rosalina* spp. | 0 | 0 | 0 | 32 | 0 | 0 | 0 | 0 | 0 | 102 | 0 | 0 | 0 | 0 | 0 | 0 | 0 | 0 | 0 | 0 | 0 | 0 |
| *Stainforthia fusiformis* | 126 | 119 | 746 | 669 | 827 | 277 | 373 | 340 | 547 | 306 | 258 | 838 | 1025 | 2029 | 402 | 541 | 1096 | 2112 | 1144 | 427 | 304 | 161 |
| *Stainforthia loeblichi* | 0 | 0 | 17 | 16 | 0 | 0 | 0 | 0 | 0 | 0 | 0 | 0 | 8 | 0 | 0 | 8 | 7 | 0 | 7 | 0 | 16 | 0 |
| Hyalin indet (round) | 0 | 0 | 9 | 0 | 0 | 0 | 0 | 0 | 68 | 68 | 15 | 14 | 15 | 0 | 0 | 0 | 7 | 14 | 14 | 0 | 0 | 2 |
| Hyalin indet (twisted) | 0 | 0 | 17 | 0 | 18 | 0 | 0 | 38 | 0 | 34 | 30 | 0 | 8 | 0 | 0 | 0 | 0 | 0 | 0 | 0 | 0 | 0 |
| Hyalin varia | 6 | 0 | 0 | 0 | 0 | 0 | 0 | 0 | 0 | 34 | 0 | 0 | 0 | 0 | 0 | 0 | 0 | 0 | 0 | 0 | 0 | 2 |
| *Adercotryma glomerata* | 13 | 44 | 206 | 127 | 35 | 0 | 14 | 0 | 0 | 0 | 0 | 0 | 15 | 0 | 0 | 0 | 0 | 0 | 0 | 0 | 0 | 0 |
| *Ammodiscus* sp. | 0 | 0 | 9 | 32 | 18 | 0 | 0 | 0 | 0 | 0 | 0 | 0 | 0 | 0 | 0 | 0 | 0 | 0 | 0 | 0 | 0 | 2 |
| *Ammoscalaria pseudospiralis* | 6 | 0 | 51 | 8 | 53 | 22 | 41 | 189 | 589 | 484 | 319 | 65 | 8 | 8 | 15 | 0 | 0 | 0 | 14 | 9 | 0 | 0 |
| *Ammotium cassis* | 1 | 0 | 0 | 80 | 18 | 0 | 0 | 0 | 0 | 0 | 0 | 0 | 0 | 0 | 0 | 0 | 0 | 0 | 0 | 0 | 0 | 0 |
| *Cribrostomoides crassimargo* | 0 | 0 | 17 | 16 | 106 | 30 | 28 | 0 | 0 | 0 | 0 | 0 | 0 | 0 | 0 | 0 | 0 | 14 | 0 | 9 | 0 | 2 |
| *Cribrostomoides subglobosum* | 0 | 2 | 0 | 0 | 0 | 0 | 0 | 19 | 0 | 0 | 0 | 0 | 0 | 0 | 0 | 0 | 7 | 0 | 0 | 0 | 0 | 0 |
| *Cribrostomoides* spp. | 0 | 0 | 206 | 207 | 317 | 45 | 69 | 19 | 103 | 170 | 46 | 116 | 62 | 38 | 44 | 16 | 21 | 28 | 14 | 26 | 16 | 2 |
| *Recurvoides* spp. | 57 | 44 | 0 | 0 | 53 | 0 | 28 | 0 | 0 | 0 | 0 | 0 | 0 | 0 | 0 | 0 | 0 | 0 | 0 | 0 | 0 | 0 |
| *Eggerelloides medius/scabrus* | 189 | 170 | 1055 | 1115 | 986 | 847 | 1133 | 4327 | 7756 | 9279 | 5696 | 3769 | 1125 | 712 | 920 | 470 | 516 | 514 | 1349 | 1325 | 793 | 223 |
| *Haplophragmoides bradyi* | 6 | 0 | 0 | 0 | 0 | 0 | 0 | 0 | 0 | 0 | 0 | 0 | 0 | 0 | 0 | 0 | 0 | 0 | 0 | 0 | 0 | 0 |
| *Lagenammina difflugiformis* | 25 | 6 | 26 | 0 | 70 | 0 | 0 | 0 | 26 | 0 | 76 | 0 | 0 | 0 | 7 | 8 | 7 | 14 | 0 | 17 | 0 | 8 |
| *Leptohalysis scotti* | 63 | 25 | 0 | 0 | 0 | 0 | 0 | 0 | 0 | 0 | 0 | 0 | 0 | 0 | 0 | 0 | 0 | 0 | 0 | 0 | 0 | 0 |
| *Miliammina fusca* | 0 | 0 | 26 | 32 | 0 | 7 | 0 | 19 | 0 | 102 | 0 | 0 | 0 | 23 | 0 | 0 | 7 | 21 | 7 | 9 | 0 | 2 |



Appendix B. Total faunas, normalized to 50 cm³

| | | | | | | | | | | | | | | | | | | | | | | |
|---|---|---|---|---|---|---|---|---|---|---|---|---|---|---|---|---|---|---|---|---|---|---|
| *Paratrochammina haynesi* | 0 | 0 | 0 | 0 | 0 | 0 | 0 | 0 | 0 | 102 | 0 | 14 | 0 | 0 | 0 | 0 | 0 | 0 | 0 | 0 | 0 | 0 |
| *Psammosphaera bowmanni* | 6 | 0 | 0 | 0 | 18 | 0 | 14 | 0 | 0 | 0 | 0 | 0 | 0 | 8 | 0 | 0 | 0 | 0 | 0 | 0 | 0 | 0 |
| *Reophax subfusiformis* | 285 | 181 | 583 | 430 | 722 | 127 | 207 | 557 | 1102 | 1198 | 440 | 173 | 139 | 106 | 153 | 39 | 29 | 56 | 92 | 60 | 32 | 27 |
| *Reophax* spp. | 0 | 0 | 0 | 0 | 0 | 0 | 14 | 0 | 0 | 0 | 0 | 0 | 0 | 0 | 0 | 0 | 0 | 0 | 0 | 0 | 0 | 0 |
| *Spiroplectammina biformis* | 19 | 50 | 343 | 207 | 282 | 30 | 138 | 0 | 0 | 0 | 0 | 0 | 62 | 83 | 22 | 47 | 43 | 42 | 35 | 0 | 0 | 20 |
| *Textularia earlandi* | 57 | 0 | 60 | 0 | 88 | 0 | 0 | 0 | 0 | 0 | 0 | 0 | 0 | 8 | 0 | 0 | 0 | 7 | 0 | 0 | 0 | 0 |
| *Textularia kattegatensis* | 0 | 6 | 0 | 0 | 0 | 0 | 0 | 0 | 0 | 0 | 0 | 0 | 0 | 0 | 0 | 0 | 0 | 0 | 0 | 0 | 0 | 0 |
| *Textularia* spp. | 0 | 0 | 26 | 0 | 0 | 7 | 0 | 0 | 0 | 0 | 0 | 0 | 0 | 0 | 0 | 0 | 0 | 0 | 0 | 0 | 0 | 0 |
| *Rotaliammina adaperta* | 0 | 0 | 0 | 32 | 53 | 22 | 41 | 0 | 34 | 68 | 46 | 14 | 8 | 23 | 15 | 0 | 29 | 21 | 21 | 9 | 8 | 12 |
| *Trochammina* spp. | 0 | 0 | 0 | 0 | 53 | 0 | 28 | 19 | 0 | 102 | 46 | 29 | 8 | 0 | 0 | 24 | 0 | 21 | 0 | 9 | 0 | 0 |
| Agglutinated varia | 6 | 19 | 137 | 0 | 0 | 0 | 0 | 76 | 0 | 0 | 0 | 0 | 100 | 114 | 139 | 78 | 136 | 77 | 92 | 0 | 104 | 31 |
| | | | | | | | | | | | | | | | | | | | | | | |
| TOTAL | 2192 | 2198 | 8472 | 7418 | 8933 | 3620 | 4304 | 11725 | 19544 | 22561 | 12015 | 8968 | 4045 | 4308 | 2511 | 2187 | 2694 | 4013 | 3963 | 2854 | 2147 | 788 |