# Peer review of "Rapid environmental responses to climate-induced hydrographic changes in"

_Biogeosciences, 2019_

## Referee Comment (RC1) · Anonymous Referee #1 · 9 Jul 2019

Review of the manuscript bg-2019-199

The manuscript bg-2019-199 entitled "Rapid environmental responses to climate-induced hydrographic changes in the Baltic Sea entrance" of the authors Laurie M. Charrieau et al., investigates the recent evolution (200 years) of benthic system in the Oresun (Baltic Sea). It consists in a multidisciplinary study based on benthic foraminifera and sedimentological parameters, supported by atmospheric long time series. Along the sedimentary record they identified five different foraminiferal zones, associated to environmental changes. Globally they attributed these environmental changes to changing in the current velocities and to anthropogenic-induced eutrophi-

cation. Overall, the manuscript is well written. It shows nicely the use of foraminiferal-based proxy to reconstruct in detail modern environments. Although the initial part could be shortened, it well introduces the problematics of the paper. The results are consistent with the applied methodologies. However, some concerns can arise relatively to the statistical interpretation and some parts of the discussions that result to be inconsistent. In light of this, I suggest the present manuscript as suitable for publication in Biogeosciences only after minor revisions. Hereafter my comments and suggestions.

General comments

The first part of the MS (Introduction and Study area) is well written. However, the information relative to the figures 2 and 3 result to be redundant for the paper. Although the data presented in both figures might represent a good background of the study area, they are not discussed in the article. I may suggest shortening this part and put the figures 2 and 3 as supplementary materials. A concern reading the paper is how the authors have identified the 5 different foraminiferal zones along the sedimentary record. The frame of the discussion is based on that. As mentioned in materials and methods, they used a constrained Cluster Analysis (CA) using the Morisita's index. The relative dendrogram based on the arithmetic average (with the UPGMA) seems to be consist, however the final attribution into three foraminiferal zones separated in 5 subzones is totally no sense. The choice of the final clusters can be made by two ways: 1) "expert judgement" and 2) statistical significance. I believe in this case the Authors' choice was based on the first one (if it is not the case the Authors should explain that). What we can clearly see from the dendrogram is that there are two main clusters: Cluster 1 including samples from 36 to 4 and Cluster 2 including samples from 1 to 2. Secondary Cluster 1 can be divided in two subclusters: 1a from 36 to 18 and 1b from 17 to 4. Consequently the interpretation of the CA (made by the Authors) is not consistent. I suggested to revise this part proposing a different interpretation of the CA or another alternative statistical analysis. In any case the discussion should be rearranged accordingly. Normally, the results of abiotic parameters are shown before

the biotic parameters because faunal distributions are dependent (or not) on them. In this study is not the case. I suggest to describe before environmental parameters and then the foraminiferal assemblages. The discussions are sometimes not persuasive. I personally respected the fact that the discussions are very detailed and sharp but sometime the data do not support your statements. In some cases these statements are contradictory. I found that some considerations are too speculative, especially concerning the human interactions. I suggest to reconsider some of them. Look into specific comments.

Specific comments

Line 11: Replace "foraminiferal" with "foraminifera". Line 23: The largest changes occurred.. in? from? 1950 Line 25: The authors may think to replace Elphidium group to Elphididae. Line 26: Replace "more sandy" to "sandier". Line 28: I am not sure in the abstract acronyms or abbreviations are accepted. Please check it. Line 31-33: I suggest to rephrase this sentence or split in twice. Line 32: get rid "species" and keep only "foraminiferal assemblage". Line 43-45: "The region is...Baltic Sea". Please add a reference for this statement. Line 70: I think you can add more recent references than Sen Gupta 1999. Line 77: Get rid "analysis". Line 76-79: The Authors may think to slightly rephrase adding "The objective of this study..." Line 84: I think the "-" between 1948 and 2013 is too long. Line 90: Replace ";" with ".". Line 109: Replace "In" with "At". Line124: Please specify in the brackets what is CTD. Line 126-129: "The CTD...bottom water". I do not see the interest of using these data, already published. In any case this part has to be moved to the section study area or results. Line 150: I suggest to get rid this sentence because you did not strictly follow Murray 2006 (very general work). In addition you explain just after the sample preparation. Line 153-154: Why 100-500 $\mu$m? In benthic foraminiferal studies the common fraction is >63 $\mu$m. Is there any specific reasons? Can the Authors justify that? Line 154-156: I think you should specify how many samples you finally have (22 right?). Line 156: Why 300 specimens? Is there any reasons of that? Line 180-181: This is not clear. From this

sentence the readers understand that you are dealing (in the first two centimetres) with living fauna. This is not the case. Please get rid this sentence or correct it. Line 187-188: I think you should mention here that you calculated the Shannon Diversity as well. Line 190: You should detail the formula of the Morisita's index as you did for the FAR. In benthic foraminiferal studies this index is not so common. Line 221: You should add the meaning of NOA. Line 240-241: Add the percentage of porcelaneous (x), hyalin (x) and agglutinated (x). Line 243-246: For this part please refer to the comments afore-mentioned. Line 347-348: I disagree with this sentence. Low foraminiferal diversity can be due to many reasons and not only salinity. Although in brackish environments (and generally in transitional environments) foraminiferal density is low, this is linked to many factors. Amongst these, for instance the fact that these environments are natu-rally stressed (rapid changing of physical-chemical parameters) is a major one. Line 352: Please add the unit for salinity. Line 352: As far as I know salinity in brackish water is 0.5-30 ‰İf it is so, the fact that you found typical brackish species (tolerant to low salinity) in this interval is definitely in the contrast with this sentence. Please clarify it. I agree with you about low oxygen conditions but affirming that "salinity was about 30" is not so persuasive. Line 359: Again here saying that low diversity is "usually" link to salinity, needs a better explanation. Line 360: "However. . . least 32". Please can you make a reference for this statement? Line 362-374: I do not see from your data how you can have evidence of pollution in this interval. The only evidence you have is that "TOC was high in this interval but not higher than in the previous zone". So how can you speculate so? Based on foraminiferal diversity and abundance? I think you should have more evidence than that. In the previous interval diversities are even lower. This part is not persuasive at all. Line 370: I think you can use a better and more recent reference than this. Line 372-374: From Table 2 R. subfusiformis is tolerant to environmental variations not to various kind of pollution. Then what does it means? A lot of foraminifera are tolerant to environmental variations. This is a very general statement and do not support this part of the discussion. Line 390-393: Sorry but I do not see how you can state this. How has the increase of organic matter been

beneficial for foraminifera? All foraminiferal species had a shutdown after 1980. Could the authors explain that or modify this statement? Line 416-418. This is totally in contrast with the statement in 390-393. You said that there was an increase of nutrients loadings after 1980 and now you state that in the same period measures were taken to reduce nutrients discharges. I think you must clarify all this part concern human impact. Not clear at all. 430: "since after" Since or after? 436: Why open ocean salinity? Elphidium group includes typical brackish species (line 350). This in the contrast with this last statement. Please clarify. Figures Figure 1: The contours have a low definition. It is possible to have higher quality picture? Figure 2-3: Look the aforementioned comments. In addition data from figure 2 have been published before by Laurie M. Charrieau et al. 2018. Figure 3 is not totally clear; it shows seasonal variations of several parameters based on uncertain measurements from 1956. It does not show any variation along the fossil record. I am sorry but I do not see how it can support the study. Figure 5: Is useful to show both relative and absolute abundance? Figure 6 The subcluster are not marked. FOR-B and FOR-C are subclusters. Figure 8-9. Why now did you invert the order of the axes? I think I could be easier for the readers to keep always the same orientation. Maybe the authors may think to add a synthetic picture with the main parameters used for the reconstruction. I think this could help the readers to better follow the discussions and the final conclusions.

List of references was not checked for completeness.

Please also note the supplement to this comment:
https://www.biogeosciences-discuss.net/bg-2019-199/bg-2019-199-RC1-supplement.pdf

---

## Referee Comment (RC2) · Anonymous Referee #2 · 10 Jul 2019

Review of Charrieau et al ' Rapid environmental response to climate-induced hydro-graphic changes in the Baltic Sea entrance.

The authors use benthic foraminifera and sediment geochemistry to reconstruct environmental changes in the Sound of the Danish Strait for the last 200 years, in relation to anthropogenic disturbances. The results suggest that large changes occurred around 1950, where the foraminiferal faunal assemblage shifted to a more diverse fauna, and sediments became more sandy. The authors relate this to a shift in current strength. Between 1870-1953 ($\sim$Industrial revolution) benthic foraminifera assemblages were low as was diversity, which may potentially be linked to input of waste waters (industrial

and domestic).

The manuscript is clear and well written. I only have a couple of questions/comments, as discussed below:

Major: It would be good to include a discussion about the sedimentology in the area. For examples, how can you be confident that the coarser sandier part in the top of the core is not part of a natural succession of a migrating bar? Were any duplicate cores takes from the wider area that show the same feature?

Is there any other data from 1870-1953 interval that provides evidence for pollution (e.g. trace metals in benthic foraminifera for examples?)

Minor: - freeze drying of sediments poses a risk of losing more fragile foraminifera, including organic walled specimens, - lines 323-330: from figure 8 it seems that there are periods with high and low VAV, but there does not seem to be a direct response within the assemblage of FOR-B2,

-could the higher accumulation rates (figure 4) be partially related to the top 10 cm being less compacted (and dense) compared with further downcore in the sediments?

---

## Author Comment (AC1) · 13 Aug 2019

General comments: - The first part of the MS (Introduction and Study area) is well written. However, the information relative to the figures 2 and 3 result to be redundant for the paper. Although the data presented in both figures might represent a good background of the study area, they are not discussed in the article. I may suggest shortening this part and put the figures 2 and 3 as supplementary materials.

Reply: We have considered the reviewer's comment, and decided to keep the Figures 2 and 3 in the paper. Indeed, the water stratification in the Öresund is rather unusual as compared to other coastal regions, and both figures give important keys to picture the

system. Figure 3 shows the seasonal variability of salinity, temperature, pH, oxygen and dissolved inorganic nitrogen in the Öresund, and Figure 2 shows the two layers stratification at the site during our specific sampling time. We think both figures are necessary to understand the complex environmental settings of the area, and to better understand the discussion that follows. We have however shortened the study site section to make it clearer.

- A concern reading the paper is how the authors have identified the 5 different foraminiferal zones along the sedimentary record. The frame of the discussion is based on that. As mentioned in materials and methods, they used a constrained Cluster Analysis (CA) using the Morisita's index. The relative dendrogram based on the arithmetic average (with the UPGMA) seems to be consist, however the final attribution into three foraminiferal zones separated in 5 subzones is totally no sense. The choice of the final clusters can be made by two ways: 1) "expert judgement" and 2) statistical significance. I believe in this case the Authors' choice was based on the first one (if it is not the case the Authors should explain that). What we can clearly see from the dendrogram is that there are two main clusters: Cluster 1 including samples from 36 to 4 and Cluster 2 including samples from 1 to 2. Secondary Cluster 1 can be divided in two subclusters: 1a from 36 to 18 and 1b from 17 to 4. Consequently the interpretation of the CA (made by the Authors) is not consistent. I suggested to revise this part proposing a different interpretation of the CA or another alternative statistical analysis. In any case the discussion should be rearranged accordingly.

Reply: The choice of final clusters was first based on statistical significance from the cluster analysis to differentiate the three main zones: FOR-A (36-18), FOR-B (17-4) and FOR-C (2-1). These three main zones are clearly displayed in the dendrogram on Figure 6. The sub-clusters were then chosen not only based on the cluster analysis, but also based on the factors from the correspondence analysis. These factors are shown on the Figure 5. Combined together, the two statistical analyses allow us to suggest 5 different sub-zones in our foraminiferal record: FOR-A1 (36-30), FOR-A2

(28-18), FOR-B1 (17-8), FOR-B2 (6-4) and FOR-C (2-1). We have now clarified in the results section how the 5 foraminiferal zones were determined in our study.

- Normally, the results of abiotic parameters are shown before the biotic parameters because faunal distributions are dependent (or not) on them. In this study is not the case. I suggest to describe before environmental parameters and then the foraminiferal assemblages.

Reply: We agree with the reviewer that abiotic parameters are often presented before the faunal data in foraminiferal papers. However, we think that foraminiferal data in our study are the most important result compared to other parameters. Moreover, we organised the results and discussion sections based on the foraminiferal zones, starting from the oldest zone FOR-A1. In this first zone, abiotic data were not available for all centimeters, and only the faunal data can be fully discussed. Thus, in our study we have chosen to present the faunal data before the environment parameters.

- The discussions are sometimes not persuasive. I personally respected the fact that the discussions are very detailed and sharp but sometime the data do not support your statements. In some cases these statements are contradictory. I found that some considerations are too speculative, especially concerning the human interactions. I suggest to reconsider some of them. Look into specific comments. Reply: We have revised the discussion part while keeping the suggestions of the reviewer in mind, and we have modified some statements. We think that the discussion is now clearer and more accurate.

Minor comments: Line 11: Replace "foraminiferal" with "foraminifera". Reply: done.

Line 23: The largest changes occurred.. in? from? 1950 Reply: This was modified.

Line 25: The authors may think to replace Elphidium group to Elphididae. Reply: We changed Elphidium group into Elphidium species in the whole text. As most of the time we precise which Elphidium species we are talking about, we think the world

Elphidiidae would be confusing for the reader.

Line 26: Replace "more sandy" to "sandier". Reply: done.

Line 28: I am not sure in the abstract acronyms or abbreviations are accepted. Please check it. Reply: The NAO index is now spelled out.

Line 31-33: I suggest to rephrase this sentence or split in twice. Reply: The sentence in the abstract was split in two shorter sentences.

Line 32: get rid "species" and keep only "foraminiferal assemblage". Reply: done.

Line 43-45: "The region is...Baltic Sea". Please add a reference for this statement. Reply: done.

Line 70: I think you can add more recent references than Sen Gupta 1999. Reply: We have added references.

Line 77: Get rid "analysis". Reply: done.

Line 76-79: The Authors may think to slightly rephrase adding "The objective of this study..." Reply: We have rephrased this sentence as suggested.

Line 84: I think the "-" between 1948 and 2013 is too long. Reply: This was modified.

Line 90: Replace ";" with ".". Reply: done.

Line 109: Replace "In" with "At". Reply: done

Line124: Please specify in the brackets what is CTD. Reply: done.

Line 126-129: "The CTD...bottom water". I do not see the interest of using these data, already published. In any case this part has to be moved to the section study area or results. Reply: This section was deleted as it is indeed already discussed in Charrieau et al (2018).

Line 150: I suggest to get rid this sentence because you did not strictly follow Murray

2006 (very general work). In addition you explain just after the sample preparation. Reply: This sentence was deleted.

Line 153-154: Why 100-500 $\mu$m? In benthic foraminiferal studies the common fraction is >63 $\mu$m. Is there any specific reasons? Can the Authors justify that? Reply: In this study, the foraminiferal data from the core tops come from previously published data in Charrieau et al. (2018). The authors chose to use the size fraction >100 $\mu$m to be able to compare their results with other studies from the same area (e.g. Conradsen et al. 1993; 1994). Thus, in this study we chose the >100 $\mu$m fraction to stay coherent with their method and to be able to use those data.

Line 154-156: I think you should specify how many samples you finally have (22 right?). Reply: 22 samples were indeed analysed, we added this information in the method section.

Line 156: Why 300 specimens? Is there any reasons of that? Reply: 300 specimens is the number of picked foraminifera recommended by Patterson and Fishbein (1989) in order to identify the species comprising 10 % or more of an assemblage. We clarified this in the method section.

Line 180-181: This is not clear. From this sentence the readers understand that you are dealing (in the first two centimetres) with living fauna. This is not the case. Please get rid this sentence or correct it. Reply: The sentence was clarified.

Line 187-188: I think you should mention here that you calculated the Shannon Diversity as well. Reply: We thank the reviewer for this suggestion and we have added a sentence about the Shannon diversity in the method section.

Line 190: You should detail the formula of the Morisita's index as you did for the FAR. In benthic foraminiferal studies this index is not so common. Reply: The Morisita index is an index of similarity recommended for quantitative data because it is not greatly affected by sample size. This index is routinely used in ecological studies, but we think

that giving the rather complex formula is not necessary in this case. Instead, we have added a reference in the method section where the Morisita index is described, for the readers willing to get more details about our statistical analysis.

Line 221: You should add the meaning of NOA. Reply: NAO (North Atlantic Oscillation) is now spelled out.

Line 240-241: Add the percentage of porcelaneous (x), hyalin (x) and agglutinated (x). Reply: done

Line 243-246: For this part please refer to the comments aforementioned. Reply: This section of the text about the cluster analysis was modified accordingly.

Line 347-348: I disagree with this sentence. Low foraminiferal diversity can be due to many reasons and not only salinity. Although in brackish environments (and generally in transitional environments) foraminiferal density is low, this is linked to many factors. Amongst these, for instance the fact that these environments are naturally stressed (rapid changing of physical-chemical parameters) is a major one. Reply: We thank the reviewer for this remark. The sentence was modified to qualify the role of salinity in foraminiferal densities.

Line 352: Please add the unit for salinity. Reply: : Even if several units can be used when discussing salinity, most of researchers recommend to not use any unit, as salinity is a conductivity ratio (see any oceanography text book). We prefer to follow that recommendation.

Line 352: As far as I know salinity in brackish water is 0.5-30 ‰İf it is so, the fact that you found typical brackish species (tolerant to low salinity) in this interval is definitely in the contrast with this sentence. Please clarify it. I agree with you about low oxygen conditions but affirming that "salinity was about 30" is not so persuasive. Reply: The fact that there were brackish water tolerant species in this zone, combined with the fact that B. marginata (typical marine species) was absent, suggest brackish water on this

period. We agree with the reviewer that the salinity of brackish water can be between 0.5 and 30. However, the salinity couldn't have been too low, as some species that do not tolerate very low salinities, like N. labradorica and S. fusiformis, were also found. The salinity was then probably around 30, which is still in the brackish conditions range. We have clarified this sentence is the discussion section.

Line 359: Again here saying that low diversity is "usually" link to salinity, needs a better explanation. Reply: We have modified this sentence.

Line 360: "However... least 32". Please can you make a reference for this statement? Reply: A reference was added.

Line 362-374: I do not see from your data how you can have evidence of pollution in this interval. The only evidence you have is that "TOC was high in this interval but not higher than in the previous zone". So how can you speculate so? Based on foraminiferal diversity and abundance? I think you should have more evidence than that. In the previous interval diversities are even lower. This part is not persuasive at all. Reply: In this part of the discussion, we want to understand the reasons for the low foraminiferal BFAR and diversity. Based on previous studies in the area, we know that pollution increased considerably in various ways during this period, which could be an explanation for water properties changes such as carbonate chemistry and pH. It is known that in general, pollution has a negative effect on foraminifera. Moreover, some pollution-tolerant species were found, such as E. medius/scabrus. This is also supported by the highest number of organic linings found during this period, compared to the rest of the record. Thus, we have considered pollution as a possible explanation for the low foraminiferal abundance and diversity during this period. We have clarified this in the text and added references in the discussion section.

Line 370: I think you can use a better and more recent reference than this. Reply: We now have added references.

Line 372-374: From Table 2 R. subfusiformis is tolerant to environmental variations not

to various kind of pollution. Then what does it means? A lot of foraminifera are tolerant to environmental variations. This is a very general statement and do not support this part of the discussion. Reply: We have added information on the Table 2.

Line 390-393: Sorry but I do not see how you can state this. How has the increase of organic matter been beneficial for foraminifera? All foraminiferal species had a shut-down after 1980. Could the authors explain that or modify this statement? Reply: This statement refers to the important changes at the beginning of this period, before ∼1980. The increase in nutrient loads may have provided food for the foraminifera. In a second time, after ∼1980, the measures taken in water treatment to reduce nutrient discharge are a possible cause of the lower growth rates. These two periods are now clarified in the text.

Line 416-418. This is totally in contrast with the statement in 390-393. You said that there was an increase of nutrients loadings after 1980 and now you state that in the same period measures were taken to reduce nutrients discharges. I think you must clarify all this part concern human impact. Not clear at all. Reply: As said above, this period can be divided in two parts. The paragraph was modified, and therefore the statement is no longer in conflict.

430: "since after" Since or after? Reply: This sentence now reads "The species is also present on the south coast of Norway since ∼ 2009".

436: Why open ocean salinity? Elphidium group includes typical brackish species (line 350). This in the contrast with this last statement. Please clarify. Reply: We agreed that the Elphidium species are often found in brackish water. However, as showed on the Table 2, they can tolerate a wide range of salinities, including open-ocean salinity. Moreover, the presence of marine species such as B. marginata suggest high salinity during this period. We added this information in the discussion section.

Figures: Figure 1: The contours have a low definition. It is possible to have higher quality picture? Reply: The quality of the Figure 1 has now been improved.

Figure 2-3: Look the aforementioned comments. In addition data from figure 2 have been published before by Laurie M. Charrieau et al. 2018. Figure 3 is not totally clear; it shows seasonal variations of several parameters based on uncertain measurements from 1956. It does not show any variation along the fossil record. I am sorry but I do not see how it can support the study. Reply: As said above, we have decided to keep the Figures 2 and 3 to help the reader to fully understand the complex system in the Öresund. The data of the Figure 3 are annual water measurements from the SMHI (Swedish Meteorological and Hydrological Institute) between 1965 and 2016, which were compiled to give the best overview of the seasonal variability in the area.

Figure 5: Is useful to show both relative and absolute abundance? Reply: The relative abundances allow us to compare the development of the major species along each zone. However, the graph alone could give the wrong impression regarding the absolute abundances, which can be extremely different between the species. We then have decided to show both relative and absolute abundances in the Figure 5.

Figure 6 The subcluster are not marked. FOR-B and FOR-C are subclusters. Reply: The dendrogram on the Figure 6 shows the three main zones FOR-A, FOR-B and FOR-C. As we explain above, the sub-clusters were then determined not only based on the cluster analysis, but also on the correspondence analysis. Thus, we chose to not report the sub-clusters on the Figure 6 to avoid confusion in the choice of foraminiferal zones. This was clarified in the results section.

Figure 8-9. Why now did you invert the order of the axes? I think I could be easier for the readers to keep always the same orientation. Reply: By reversing the axis on Figure 9 (C), it is easier to visualize the trends of BFAR, sand content and UAV in the Kattegat, while keeping the same information than on Figure 8.

Maybe the authors may think to add a synthetic picture with the main parameters used for the reconstruction. I think this could help the readers to better follow the discussions and the final conclusions. Reply: In Figure 9, the BFAR represents the foraminifera,

the sand content represents the sedimentology, and they are both compared with the water currents, wind model, and NAO index. We think that the Figure 9 is already a good synthesis of the paper, and it can be used by the reader to follow the key changes in our reconstruction of the environmental changes in the area.

We would like to thank Reviewer 1 for the insightful and helpful comments that we think have significantly improved our manuscript.

Kind regards,

Laurie M. Charrieau, on behalf of the authors: Karl Ljung, Frederik Schenk, Ute Daewel, Emma Kritzberg and Helena L. Filipsson.

———————————————————

Figure 1. Map of the studied area. The star shows the focused station of this study. General water circulation: main surface currents (black arrows) and main deep currents (grey arrows). GB: Great Belt; LB: Little Belt; AW: Atlantic Water; CNSW: Central North Sea Water; JCW: Jutland Coastal Water; NCC: Norwegian Coastal Current; BW: Baltic Water. Insert source: © BSHC.

**Fig. 1.** Map of the studied area.

[Figure]

Figure 2. CTD profiles of temperature, salinity, pH and dissolved oxygen concentration in the

water column for the DV-1 station (modified from Charrieau et al. 2018).

**Fig. 2.** CTD profiles of temperature, salinity, pH and dissolved oxygen concentration in the water column for the DV-1 station (modified from Charrieau et al. 2018).

[Figure]

Figure 3. Seasonal variability of salinity, temperature, pH and dissolved inorganic nitrogen (DIN)
concentration at the surface water (light grey), and seasonal variability of salinity, temperature, pH and
dissolved oxygen concentration at the bottom water (40-50 m) (dark grey) of the Öresund. The data
were measured between 1965 and 2016 by the SMHI (Swedish Meteorological and Hydrological
Institute) at the station W LANDSKRONA. The number of measurements is indicated for each month.

**Fig. 3.** Seasonal variability of salinity, temperature, pH and dissolved inorganic nitrogen (DIN)
concentration at the surface water (light grey), and seasonal variability of salinity, temperature,
pH and DIN

[Figure]

Figure 5. A) Relative abundances (%) of the foraminiferal major species (>5 %), benthic foraminiferal accumulation rate (BFAR, specimens.cm$^{-2}$.yr$^{-1}$), Shannon index, organic linings (specimens.cm$^{-2}$.yr$^{-1}$) and factors from the correspondence analysis. B) Benthic foraminiferal accumulation rates (specimens.cm$^{-2}$.yr$^{-1}$) of the major species (>5%), BFAR (specimens.cm$^{-2}$.yr$^{-1}$), Shannon index, organic linings (specimens.cm$^{-2}$.yr$^{-1}$) and factors from the correspondence analysis. Foraminiferal zones based on cluster and correspondence analyses. Note the different scale on the x axes.

**Fig. 4.** A) Relative abundances (%) of the foraminiferal major species (>5 %), B) Benthic foraminiferal accumulation rates

Similarity

0.50 0.55 0.60 0.65 0.70 0.75 0.80 0.85 0.90 0.95

Figure 6. Dendrogram produced by the cluster analysis based on the Morisita index and the

UPGMA clustering method.

**Fig. 5.** Dendrogram produced by the cluster analysis

[Figure]

Figure 8. South-North flow (VAV) in the Öresund (dark line) and West-East flow (UAV) in
the Kattegat (light line) between 1950 and 2013. Foraminiferal zones indicated.

**Fig. 6.** South-North flow (VAV) in the Öresund (dark line) and West-East flow (UAV) in the
Kattegat (light line)

Figure 9. A) NAO index for boreal winter (December to March), data from Jones et al. (1997).

B) Variations of near-surface (10 m) wind conditions (October to March), data from Schenk

and Zorita (2012). Both NAO index and wind speed data are normalized on the period 1850-

2008 and show running decadal means. C) BFAR, percentage of sand fraction and West-East

flow (UAV) in the Kattegat. Foraminiferal zones indicated.

**Fig. 7.** A) NAO index for boreal winter (December to March), B) Variations of near-surface (10 m) wind conditions, C) BFAR, percentage of sand fraction and West-East flow (UAV) in the Kattegat.

---

## Author Comment (AC2) · 13 Aug 2019

General comments: It would be good to include a discussion about the sedimentology in the area. For examples, how can you be confident that the coarser sandier part in the top of the core is not part of a natural succession of a migrating bar? Were any duplicate cores takes from the wider area that show the same feature? Is there any other data from 1870-1953 interval that provides evidence for pollution (e.g. trace metals in benthic foraminifera for examples?)

Reply: It is challenging to obtain sediment cores in the Öresund, due to limited sediment deposition areas and high current velocities. Therefore, cores from the wider

area are not available for comparison with our core. As for the migrating bar, we do not have any evidence suggesting such phenomena in the area. We have now modified the discussion and added references to support the pollution suggestion during the period 1870-1953.

Minor comments: - freeze drying of sediments poses a risk of losing more fragile foraminifera, including organic walled specimens, Reply: We agree with the reviewer that freeze-drying sediment can cause loss of some of the most fragile specimens. However, the freeze-drying process was probably not a major problem for the general faunal distribution. Moreover, we found organic linings of foraminifera in our sediment. These organic linings were found undamaged, even though they could easily be broken by manipulating them with a brush. Thus, we think that the risk of losing fragile forms was minimum in our samples.

- lines 323-330: from figure 8 it seems that there are periods with high and low VAV, but there does not seem to be a direct response within the assemblage of FOR-B2, Reply: In our interpretation, the UAV was one of the most important factors to explain the foraminiferal assemblage, as showed in Figure 9. However, the resolution of our sub-sampling for foraminifera and sedimentological parameters limit the possibility to accurately resolve very short events, such as those in the topmost part of the VAV reconstruction.

- could the higher accumulation rates (figure 4) be partially related to the top 10 cm being less compacted (and dense) compared with further downcore in the sediments? Reply: We agree with the reviewer that less compact sediment in the top part of the sediment sequence is contributing to the higher sedimentation rate on this section.

We would like to thank Reviewer 2 for the insightful and helpful comments that we think have significantly improved our manuscript.

Kind regards,

Laurie M. Charrieau, on behalf of the authors: Karl Ljung, Frederik Schenk, Ute Daewel, Emma Kritzberg and Helena L. Filipsson.

[Figure]

Figure 4. Age-depth calibration for the sediment sequence from the Öresund (DV-1). A) Total and supported $^{210}$Pb activity. B) Unsupported $^{210}$Pb activity and the associated age-model. C) $^{137}$Cs activity. The peak corresponds to the Chernobyl reactor accident in 1986. D) Age-depth model for the whole sediment sequence based on $^{210}$Pb dates and calculated sediment accumulation rates (SAR).

**Fig. 1.** Age-depth calibration for the sediment sequence from the Öresund (DV-1).

[Figure]

Figure 8. South-North flow (VAV) in the Öresund (dark line) and West-East flow (UAV) in the Kattegat (light line) between 1950 and 2013. Foraminiferal zones indicated.

**Fig. 2.** South-North flow (VAV) in the Öresund (dark line) and West-East flow (UAV) in the Kattegat (light line) between 1950 and 2013.

[Figure]

Figure 9. A) NAO index for boreal winter (December to March), data from Jones et al. (1997).
B) Variations of near-surface (10 m) wind conditions (October to March), data from Schenk
and Zorita (2012). Both NAO index and wind speed data are normalized on the period 1850-
2008 and show running decadal means. C) BFAR, percentage of sand fraction and West-East
flow (UAV) in the Kattegat. Foraminiferal zones indicated.

**Fig. 3.** A) NAO index for boreal winter, B) Variations of near-surface (10 m) wind conditions, C)
BFAR, percentage of sand fraction and West-East flow (UAV) in the Kattegat